# A linear and circular dual-conformation noncoding RNA involved in oxidative stress tolerance in *Bacillus altitudinis*

Ting-Ting He [1,2,3], Yun-Fan Xu [1,3], Xiang Li[1], Xia Wang[1], Jie-Yu Li[1], Dan Ou-Yang[1], Han-Sen Cheng[1], Hao-Yang Li[1], Jia Qin[1], Yu Huang[1] & Hai-Yan Wang [1] ✉

Circular RNAs have been extensively studied in eukaryotes, but their presence and/or biological functionality in bacteria are unclear. Here, we show that a regulatory noncoding RNA (DucS) exists in both linear and circular conformation in *Bacillus altitudinis*. The linear forms promote *B. altitudinis* tolerance to $H_2O_2$ stress, partly through increased translation of a stress-responsive gene, *htrA*. The 3′ end sequences of the linear forms are crucial for RNA circularization, and formation of circular forms can decrease the levels of the regulatory linear cognates. Bioinformatic analysis of available RNA-seq datasets from 30 bacterial species revealed multiple circular RNA candidates, distinct from DucS, for all the examined species. Experiments testing for the presence of selected circular RNA candidates in four species successfully validated 7 out of 9 candidates from *B. altitudinis* and 4 out of 5 candidates from *Bacillus paralicheniformis*; However, none of the candidates tested for *Bacillus subtilis* and *Escherichia coli* were detected. Our work identifies a dual-conformation regulatory RNA in *B. altitutidinis*, and indicates that circular RNAs exist in diverse bacteria. However, circularization of specific RNAs does not seem to be conserved across species, and the circularization mechanisms and biological functionality of the circular forms remain unclear.

In the last 20 years, bacterial small RNAs (sRNAs) have gone from biological curiosity to being recognized as a major class of post-transcriptional regulators[1], playing roles in diverse cellular processes, especially under specific growth and stress conditions. The major working mode of these sRNAs is roughly analogous to that of eukaryotic microRNAs, mediated by short base pairing to regulate multiple target mRNAs, which can be as many as ~1% of all mRNAs for a single sRNA[2]. This base pairing alters mRNA stability or translation, thus resulting in the inhibition or activation of target gene expression. Nonetheless, some sRNAs act through direct interactions with regulatory proteins, e.g., the RNA component of the ubiquitous signal recognition particle (SRP) and sRNA CsrB/CsrC which sequester the small protein CsrA[3–5]. To date, all

bacterial noncoding RNAs have been reported to function in linear forms.

Circular RNAs, a class of noncoding RNAs first found in eukaryotes more than 40 years ago, have recently been shown to exert biological functions by acting as decoys, transporters, scaffolds, templates for translation, etc.[6,7]. Circular RNAs have been shown to not only be involved in a variety of human diseases but also participate in plant stress responses[8]. Circular RNAs are also discovered in archaea, whose gene expression are more similar to eukaryotes. In archaea, circular RNAs can be formed from excised tRNA-introns and rRNA processing intermediates, as well as a few functional noncoding RNAs, including box C/D RNA, RNase P, and SRP RNA[9–12]. In bacteria, the identified circular RNAs are all from the autocatalytic Group I and II introns, a

[1]Key Laboratory of Bio-Resource and Eco-Environment of Ministry of Education, College of Life Sciences, Sichuan University, Chengdu, Sichuan, China. [2]Department of Thoracic Surgery, West China Hospital, Sichuan University, Chengdu, China. [3]These authors contributed equally: Ting-Ting He, Yun-Fan Xu. ✉e-mail: hayawang@scu.edu.cn

class of self-cleaving ribozymes[13,14], except for a predicted circular tmRNA, whose presence was supported only by bioinformatic analysis[15].

Bacillus strains are important industrial enzyme producers[16], and many commercial enzymes, including proteases, amylases, xylanases, and lipases, have been characterized and purified from different Bacillus species[17–20]. To explore posttranscriptional regulation in Bacillus strains, we previously identified a set of sRNAs in Bacillus altitudinis SCU11 (initially classified as Bacillus pumilus[21]), a productive strain of extracellular alkaline protease, through an RNA-seq-based approach and subsequent validation by northern blotting[22]. We noticed that one actively transcribed novel sRNA gave rise to multiple transcripts, some with abnormal sizes. In this work, we report that this sRNA is a dual-conformation small noncoding RNA (DucS) that is found as linear and circular forms simultaneously, and confirm that DucS plays a positive regulatory role in the oxidative stress tolerance of B. altitudinis. Circular RNAs may represent a new area of bacterial noncoding RNA research.

## Results

### DucS produces abnormally sized transcripts

In previous work, a novel sRNA, Bpsr139, renamed DucS here, was identified in B. altitudinis SCU11 with a predicted size of 225 nucleotides (nt) according to transcriptome analysis and northern blotting verification[22]. DucS is encoded by the intergenic region of the B. altitudinis chromosome that exists only in several closely related species, including B. pumilus, B. cellulasensis, B. safensis, and B. xiamenensis (Supplementary Fig. 1). A DucS knockout strain ($\Delta ducS$) was constructed through the CRISPR–Cas9 system, and the complementation strain was established by introducing plasmid-borne DucS into the $\Delta ducS$ strain. Northern blotting assays revealed that both the wild-type (WT) and complementation strains produced multiple unexpected bands on denaturing urea-polyacrylamide gel electrophoresis (Fig. 1a). The expression patterns of DucS across different growth phases were further examined (Fig. 1b). In general, there were roughly four prominent and constant bands among different repeats, denoted as bands 1–4 according to migration distance from top to bottom, while other bands showing lower abundance even in the complementation strain were not marked and subsequently followed. According to the time-course expression patterns, band 1 and band 2 appeared at the mid-logarithmic phase (8 h) and increased gradually thereafter, while band 3 was enriched in the early logarithmic phase (4 h), and band 4 showed a decreasing trend with growth.

To elucidate the production of different transcripts of DucS, the accurate size of each band needed to be clarified. The total RNA was first digested with 5′ monophosphate-dependent exonuclease (TEX) to degrade processed transcripts but spare primary transcripts harboring 5′ triphosphates. Then, the treated RNA was used as a template for the cRACE assay (an improved 5′ and 3′ rapid amplification of cDNA ends from circularized RNA)[23] to identify the transcriptional start site (TSS) and transcriptional terminator site (TTS) of DucS. The sequencing results showed that the TSS of DucS was uniquely mapped to $G_{946314}$ of the B. altitudinis SCU11 genome, whereas the TTS was variable. The most prevalent TTS and the longest TTS were mapped to $A_{946512}$ (named $TTS_{1st}$) and $T_{946545}$ (named $TTS_{2nd}$), respectively (Fig. 1c, d). Based on this, the sizes of the DucS transcripts were 232 nt and 199 nt, which roughly corresponded to band 3 and band 4. The failure to identify the TSS or TTS corresponding to slowly migrating transcripts confused us, where did they come from? To study whether other TSSs existed, the expression of DucS under the control of five sequentially deleted promoters, $P_0$, $P_{35}$, $P_{74}$, $P_{120}$, and $P_{300}$ (the subscript numbers represent the length of the upstream region corresponding to TSS $G_{946314}$), was evaluated in the $\Delta ducS$ strain (Supplementary Fig. 2a). Except for $P_0$, all of the other four fragments could initiate the transcription of DucS, even though $P_{35}$ displayed a lower efficiency.

Considering that $P_{35}$ could produce all four bands of DucS (Supplementary Fig. 2b) but $P_0$ could not, we deduced that the slowly migrating transcripts (band 1 and band 2) should be initiated at the same TSS. Then, we investigated whether they were generated by terminating at variable sites, such as through read-through. To test this hypothesis, oligonucleotide probes corresponding to downstream regions of TSS and two TTSs, denoted olig1, 2, and 3, were used for northern blotting analysis (Fig. 1c, e). As a positive control, the internal oligonucleotide probe oligCK was used as it could detect all four bands. For band 3 and band 4, which were enriched in the 4 h RNA sample, olig1 hybridized with band 3&4 and olig2 only with band 3, confirming that band 3&4 initiated at the same TSS and terminated at different TTSs. Because band 1 and band 2 appeared in the 8 h sample, olig1 hybridized only with band 1, indicating that band 1 harbored a 5′ end similar to that of bands 3 and 4, while band 2 lacked this region. Considering that neither band 1 nor band 2 could be detected by olig2 or olig3 (Supplementary Fig. 2c), we speculated that neither contained the terminator and its downstream sequence. Based on the above results, the sizes of band 1 and band 2 were inferred to be less than that of band 3 (232 nt). The next question was why do they shift so slowly in electrophoresis?

### DucS may form circular RNAs

To explore the production of band 1 and band 2, we reanalyzed all the raw reads of the primary RNA-seq database. We noticed that some raw reads mapped unexpectedly to the DucS region of the reference genome in a permuted, chiastic order, which is a hallmark of circular RNA (Fig. 2a). Such raw reads (permuted reads) are often discarded during routine transcriptomic analysis due to failure to align perfectly to the reference genome. A careful analysis of the junction sites of permuted reads revealed that the 3′ termini of all junctions were fixed to $T_{946513}$, the next nucleotide after $TTS_{1st}$ $A_{946512}$, whereas the 5′ termini were variable. Most 5′ termini of the junctions corresponded to TSS $G_{946314}$ (named the S1 junction), and a small fraction was located in the region approximately 26–35 nt downstream of the TSS (named the S2 junction) (Fig. 1c). Accompanying bacterial growth, the proportion of permuted reads increased, and the S1 junction reads were always the majority (Fig. 2b). In view of these findings, we speculated that DucS could produce circular (band 1 and band 2) as well as linear (band 3 and band 4) RNAs, and two circular RNAs (200 nt and 165–174 nt) were formed through S1 or S2 circularization junctions, respectively. According to some studies, circular RNAs migrated more slowly than linear RNAs of the same length on denaturing PAGE[24–26], which supported our proposal. Taken together, these preliminary findings suggested that DucS could form circular RNAs.

### Experimental validation of DucS circular RNAs

Reverse transcription (RT) PCR was first applied to validate the circular form of DucS by using two sets of primers. The divergent primers were used to amplify the circular RNA only, while the convergent primers were used as positive controls to amplify both linear and circular forms (Fig. 2c, left). RNase R-treated total RNA was used for RT–PCR, since circular RNAs were more resistant to RNase R digestion, whereas linear RNAs were sensitive[27,28]. While the convergent primers amplified PCR products of the expected size, the divergent primers also generated clear products with RNase R-treated samples; meanwhile, a linear reference mRNA could not be amplified in this system (Fig. 2c, right). The multiple-sized products in both systems might be caused by different-sized circular RNAs or stemmed from multiple rounds of RT around a circular RNA template[11]. Sequencing the RT–PCR products from the divergent primers revealed two circularization junctions (Fig. 2d), which were consistent with transcriptomic permuted reads. Second, RNase R-treated RNA was used for northern blotting, which is regarded as the gold-standard technique to discriminate between circular and linear RNAs[6,29,30]. The results showed that the intensity of

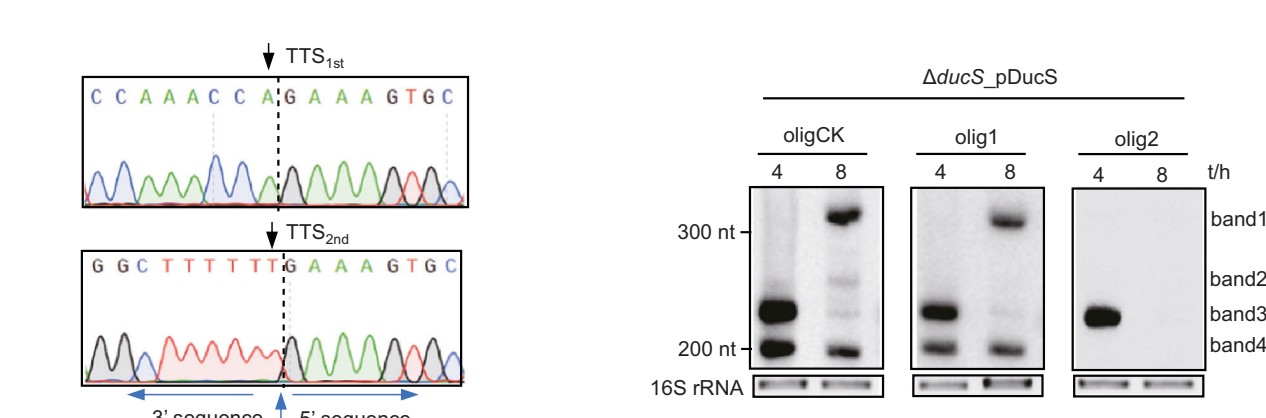

bands 1 and 2 remained almost unchanged after RNase R treatment, while that of bands 3 and 4 decreased by 67% and 74%, respectively (Fig. 2e, left). Third, the circularization junctions of DucS were detected by two junction-spanning probes, circS1 and circS2, complementary to 15 nucleotides on each side. Northern blots showed that circS1 hybridized with band 1 only and circS2 hybridized with band 2 only (Fig. 2e). Collectively, these data indicated that band 1 RNA circularized through the S1 junction and band 2 through the S2 junction, and consequently, we renamed bands 1–4 as C1, C2, L1, and L2, corresponding to circular (C) and linear (L) RNAs, respectively (Fig. 2f).

## Identification of critical sequences for DucS circularization

No circular RNA was reported before in bacteria except for self-spliced and circularized Group I or II introns[13,31,32]. No classical Group II intron

**Fig. 1 | DucS produces multiple transcripts. a**, **b** Northern blotting to detect DucS expression in *B. altitudinis* SCU11 (wild-type, WT) and its derivative Δ*ducS* carrying either the empty vector (pEV) or a plasmid expressing DucS RNA (pDucS). Probe DucS (200 nt) corresponds to almost the full length of DucS. Total RNA was extracted from the indicated strains from cultures at 8 h (**a**) or 4, 8, 12, and 24 h (**b**). **c** Genomic context of DucS. The sequence of DucS (shown in red) and its 60 nt upstream and 68 nt downstream regions are shown below. The black boxes mark the RNA circularization junctions identified in the subsequent experiments. Four oligonucleotide probes (oligCK and olig1, 2, 3) are underlined. The transcriptional

start site (TSS) and two transcriptional termination sites (TTS$_{1st}$ and TTS$_{2nd}$) (arrows) of DucS were determined by cRACE assay, whose sequencing visualization is shown in (**d**). **e** Northern blotting to clarify the four transcripts of DucS by oligonucleotide probes. For northern blotting performed in this study, unless otherwise indicated, 5 μg RNA per lane was loaded for strains harboring chromosomal DucS, and 2 μg RNA per lane was loaded for strains carrying plasmid-borne DucS. EB-stained 16S rRNA was used as a loading control. Data are representative of at least two independent experiments (**a**, **b**, **e**). Source data are provided as a Source data file.

structure was found in the DucS sequence through conserved Group II intron prediction (http://webapps2.ucalgary.ca/~groupii/index.html#). The in vitro self-splicing assay suggested that DucS could not perform self-splicing and give rise to circular RNA in vitro (Supplementary Fig. 3). Thus, circular forms of DucS may be produced through other mechanisms. To reveal the critical sequence for circularization, a series of mutants at the 5′ or 3′ end of DucS were constructed (Fig. 3a). First, the first 8 nt (DucSΔ8) or 18 nt (DucSΔ18) at the 5′-end, including TSS G$_{946314}$, were deleted. The results showed that circular RNA formation of DucSΔ18 was not affected but had a smaller size (Fig. 3b, c). However, mutant DucSΔ8 was hardly transcribed (Fig. 3b), and the substitution of the first nucleotide (T) of DucSΔ8 with other nucleotides could give rise to transcripts (Supplementary Fig. 4a). RNase R treatment followed by RT–PCR assays confirmed that both DucSΔ18 and DucSΔ8A (the first T was changed to A) could produce circular RNAs through the S2 junction (Supplementary Fig. 4b). Meanwhile, looking at the ratio of circular to linear RNAs, it seemed that the first 18 nt deletion increased the circularization efficiency (Fig. 3c).

Considering that the 3′ terminus of circularization junctions S1 and S2 was fixed to T$_{946513}$, we hypothesized that T$_{946513}$ might be important for circularization. Then, three point mutations were constructed by changing T$_{946513}$ into C (DucSM$_C$), A (DucSM$_A$), and G (DucSM$_G$) (Fig. 3a). To our surprise, these three mutants failed to produce circular RNAs, accompanied by an increase in linear RNA level (Fig. 3d). Furthermore, strains with three deletion mutations around T$_{946513}$, DucSΔ10-1, DucSΔ10-2, and DucSΔ29, were constructed to test if flanking sequences would affect circular RNA formation (Fig. 3a). DucSΔ10-1 could interfere with circularization, but the other two mutations did not (Fig. 3e, Supplementary Fig. 4c). Altogether, these results indicated that the 3′ sequences of DucS, including T$_{946513}$ and its upstream 4–13 nt, are critical to produce circular RNA. Thereafter, DucSM$_C$ was used as a circular RNA deficiency mutant for subsequent functional characterization of this sRNA.

## DucS promotes oxidative stress tolerance through its linear RNAs

In pursuit of the biological role of DucS, screening and identifying its target mRNAs were the priority. The CopraRNA[33], in combination with GLASSgo[34], was applied to target prediction, and the top 5 candidates (Supplementary Data 1) were used for electrophoretic mobility shift assays (EMSAs) to test the interactions between DucS and candidate RNAs. The results showed that the candidates *htrA*, *pbp4b*, and *yaaA* RNAs could bind with DucS RNA, but *yerD* and *yfkJ* could not (Fig. 4a, Supplementary Fig. 5). Since *htrA* encodes a serine protease, which was reported to be involved in temperature, osmotic pressure and the oxidative stress response in some gram-positive bacteria[35–37], we investigated the in vivo regulation of *htrA* by DucS by using superfolder green fluorescent protein (sfGFP) as a reporter. The assay was based on the co-expression of a translational fusion of the 5′ region of *htrA* mRNA and *sfgfp* (*htrA::sfgfp*) controlled by a constitutive promoter with or without DucS under its native promoter in the Δ*ducS* strain. The *yerD::sfgfp* fusion was used as a negative control since no interaction was detected between *yerD* and DucS in the EMSA. While the expression of DucS did not affect *yerD* fusion (Supplementary Fig. 6a), it increased the fluorescence level of the *htrA::sfgfp* fusion

(Fig. 4b). RT–qPCR results suggested that DucS did not affect the level of *htrA::sfgfp* mRNA (Supplementary Fig. 6b). Afterward, we inserted a C-terminal 3xFLAG tag into the plasmid-borne *htrA* gene (we also tried to insert the tag into the chromosomal *htrA* gene but failed) to test whether DucS affected the level of HtrA protein. Western blotting revealed that more HtrA protein appeared in the presence of DucS (Fig. 4c). The results indicated that DucS could positively regulate the translation of *htrA* mRNA. Moreover, the circular RNA deficiency mutant DucSM$_C$ showed increased fluorescence of the HtrA::sfGFP fusion (Fig. 4d), suggesting that DucS linear RNAs played a major role in regulating the target *htrA*.

Considering that DucS could positively regulate *htrA* and that *htrA* was reported to be involved in the stress response in some bacteria, we wondered if DucS played roles in *B. altitudinis* under stress conditions. We compared the growth of DucS wild-type, knockout and complementation strains after challenge with different stresses. Upon 1.5 mM H$_2$O$_2$ exposure, the Δ*ducS* strain displayed significant growth inhibition in comparison with the wild type, while the Δ*htrA* strain was only moderately inhibited, suggesting that DucS is involved in oxidative stress tolerance by regulating multiple genes in addition to *htrA* (Fig. 4e). Since strains harboring DucS or DucSM$_C$ grew similarly under H$_2$O$_2$ exposure (Fig. 4e), we speculated that circular forms of DucS might not be involved in this stress. Therefore, the expression pattern of DucS in WT under H$_2$O$_2$ exposure was detected by northern blotting, and most notably, the level of linear RNAs after 4 h of treatment was increased to some extent compared to the untreated control (Supplementary Fig. 7). The results suggested that DucS could promote the H$_2$O$_2$ stress tolerance of *B. altitudinis*, mainly through its linear RNAs.

According to in silico analysis, DucS regulated *htrA* by binding the UTR distal sequence of *htrA* mRNA, thus relieving the ribosome binding site (RBS) to increase translation (Supplementary Fig. 8a). Then, we constructed several point mutations of DucS (M$_1$, M$_2$, M$_3$ and M$_4$) to investigate their interactions with *htrA*. As the results showed, although mutations at M$_1$ and M$_4$ did not alter the regulation of DucS, mutations at two G-rich regions (M$_2$ and M$_3$) attenuated the activation of the *htrA::sfgfp* fusion, and the M$_2$/M$_3$ double mutant abolished the regulation (Supplementary Fig. 8b). Further experiments confirmed that DucS regulated oxidative stress tolerance similarly through two G-rich regions (Supplementary Fig. 8c). These data indicate that the two G-rich regions of DucS were critical for regulating *htrA* and oxidative stress tolerance.

## Circular RNAs affect the levels of their linear cognates

Since DucS regulates target *htrA* and the oxidative stress response mainly through its linear RNAs, we wondered if circular RNAs had any special roles. Comparing the expression patterns of DucS and circular RNA deficiency mutants DucSM$_{C/A/G}$ at different growth phases, we noticed that the L1 level in the WT strain decreased significantly along with growth, while L1 was maintained at a constant high level throughout the growth phases in the three circular RNA deficiency mutants (Fig. 3d). We proposed two hypotheses for this phenomenon. One is that circular RNAs are converted only from L1 since L2 is absent from the 3′ junction site; thus, the formation of circular RNAs in WT inevitably leads to a reduction in L1 level. Conversely, the deficiency of

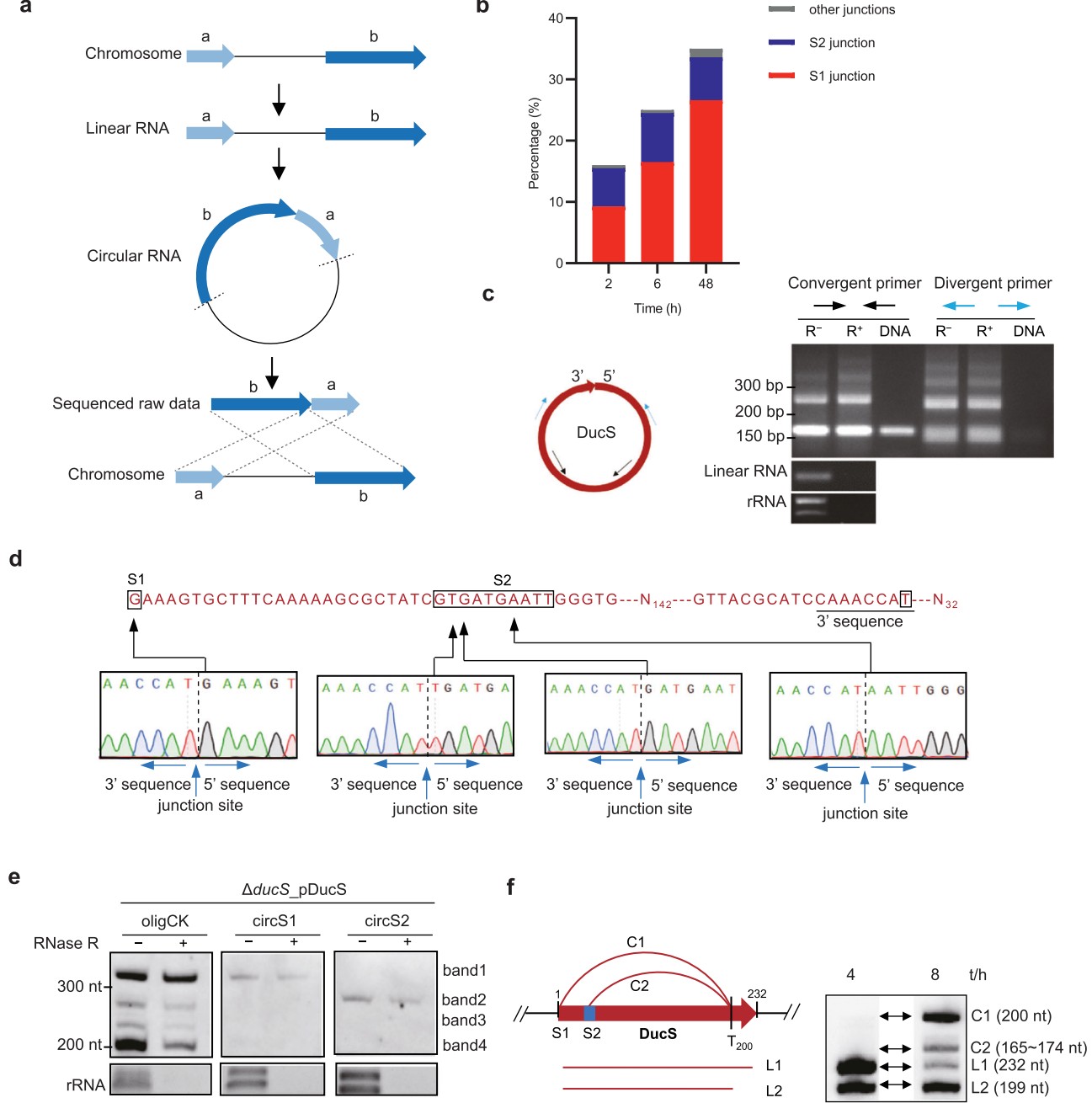

**Fig. 2 | Experimental validation of DucS circular RNAs. a** The working principal diagram for the identification of circular RNA products in RNA-seq data. Read mapping to the reference DNA in a permuted, chiastic order is a hallmark of circular RNA. **b** Percentage of the reads supporting different circularization junctions compared to the total reads aligned to the chromosomal DucS region. The locations of the S1 and S2 junctions are shown in Fig. 1c. Other junctions represent those 5′ termini that are different from S1 and S2. **c** RT–PCR verification of circular RNAs of DucS. Left: Schematic representation of primers for circular RNA verification. Black and blue arrows are convergent and divergent primer pairs, respectively. Right: RT–PCR results. cDNA templates were obtained by reverse transcription of RNA (WT, 8 h) pretreated with or without RNase R (R⁺ or R⁻). Chromosomal DNA and linear

RNA (*yaaA* mRNA) were used as a negative control. EB-stained rRNA acted as RNase R digestion controls. Similar results were obtained in two biologically independent experiments. **d** Sequencing the RT–PCR products from divergent primers. A partial sequence of DucS is shown above. $N_{142}$ and $N_{32}$ represent 142 and 32 nucleotides not shown in the sequence. Black arrows indicate different 5′ end nucleotides of circularization junctions. **e** Circular RNAs of DucS are more resistant to RNase R treatment and can be detected by junction-spanning probes. Probes circS1 and circS2 correspond to S1 and S2 junctions, respectively. Similar results were obtained in two biologically independent experiments. **f** Schematic diagram of the production of DucS transcripts and their new designations. L1 and L2 are linear RNAs. C1 and C2 are circular RNAs. Source data are provided as a Source data file.

circular RNAs in mutants results in a sustained high level of L1. The other hypothesis is that the occurrence of circular RNAs may have some effect on the stability of linear forms. To evaluate this hypothesis, we measured the half-lives of linear RNAs by adding the transcription inhibitor rifampin to Δ*ducS* strains expressing either DucS or DucSM_C and harvested samples at different time points up to 32 min.

Northern blotting showed that the presence of circular RNAs in the DucS complementation strain was accompanied by reduced half-lives of L1 and L2 (6 h *vs.* 3 h), and conversely, the circular RNA deficiency mutant possessed very stable linear RNAs, independent of the two growth phases (Fig. 4f). These phenomena suggested that circular forms of DucS could decrease the levels of their linear cognates.

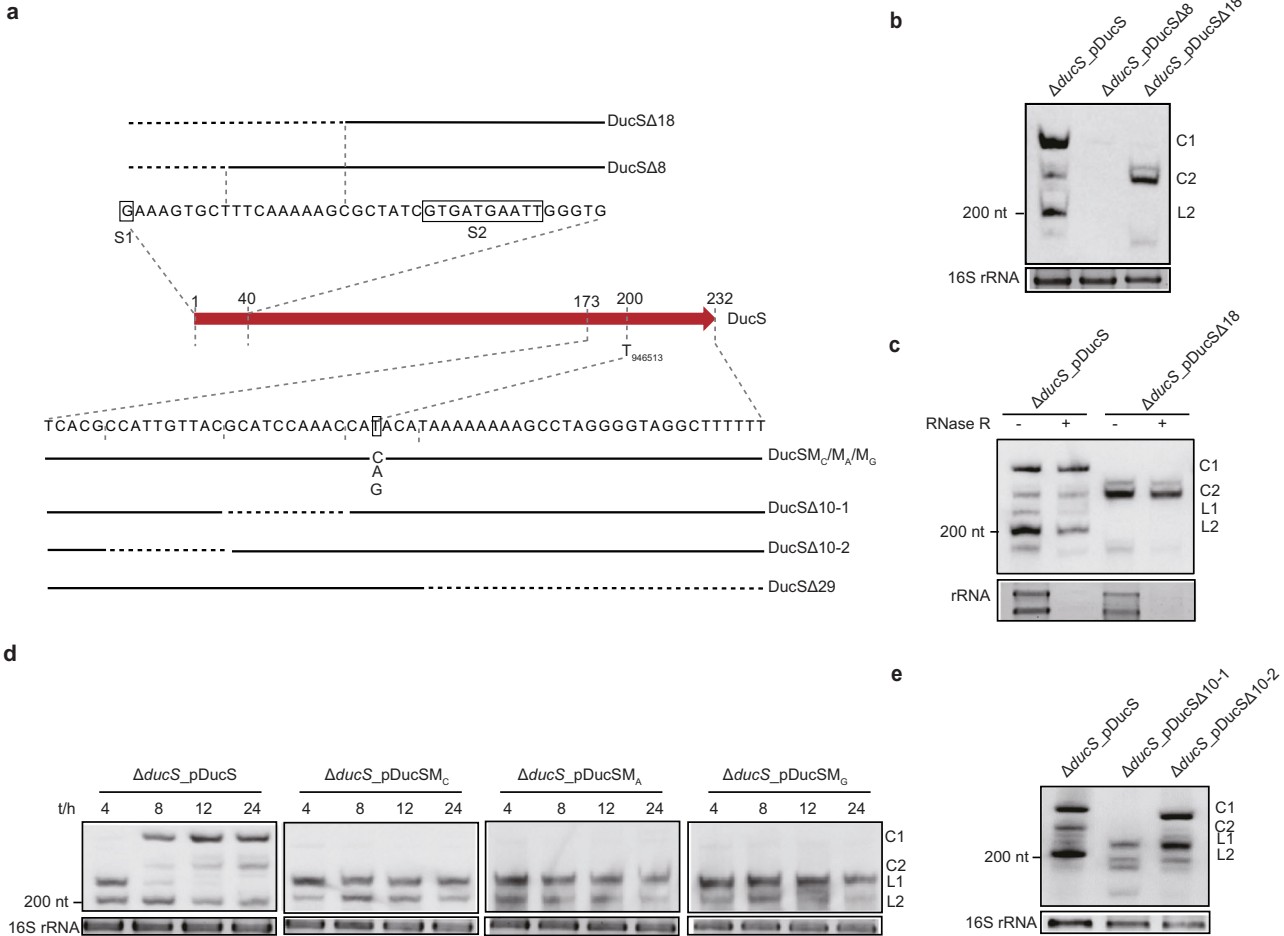

**Fig. 3 | Identification of critical sequences for DucS circularization. a** Schematic diagram of DucS variants with the indicated mutations. The solid red arrow represents DucS, and its 5′- and 3′-end sequences are shown above and below the arrow. Deletion mutations at the 5′-end of DucS (upper) and deletion or point mutations at the 3′-end (lower) were constructed on plasmid-borne DucS. The black horizontal dashed line represents deleted nucleotides. **b–e** Northern blotting to evaluate the effects of different mutations on the formation of DucS circular RNAs. Total RNA was extracted from the indicated strains from cultures at 8 h (**b**, **c**, **e**) or 4, 8, 12, and 24 h (**d**). Some samples (**c**) were treated with RNase R by using rRNA as digestion controls. Probe DucS was used for Northern blotting. EB-stained 16 S rRNA was used as a loading control. Data are representative of at least two independent experiments (**b–e**). Source data are provided as a Source data file.

## Circular RNAs also exist in other bacteria

In view of the findings that DucS could generate circular RNAs in *B. altitudinis*, we speculated that this should not be a unique phenomenon. Consequently, we sought to test whether other circular RNAs existed in *B. altitudinis*, as well as if circular RNAs could be generated in other bacteria. We analyzed published RNA-seq data from 30 bacteria by focusing on permuted reads, and the chromosomal locus mapped by more than 5 permuted reads was regarded as a circular RNA candidate. All tested strains were predicted to contain circular RNA candidates, and the candidate number of each strain ranged from a few to dozens (Fig. 5a, Supplementary Data 2). A majority of candidates were located in gene regions, and a minority were located in nonannotated RNAs and known noncoding RNAs excluding rRNA loci, since rRNAs are usually depleted before RNA-seq. In the case of each candidate, most of them possessed permuted reads less than 100, whereas a few possessed hundreds to thousands of permuted reads, which occurred mainly in *B. altitudinis*, *B. licheniformis*, *B. paralicheniformis* and *Helicobacter pylori* (Supplementary Fig. 9a). Based on circular RNA identification in archaea, permuted reads with circularization points shifted by 1–3 nucleotides were clustered into one junction to avoid redundancy[9]. Nevertheless, some candidates were mapped by distinctly different permuted reads and gave rise to several circular RNA isoforms, similar to alternative circularization in eukaryotes[38,39]. To characterize the expression pattern of circular RNA candidates, we analyzed the time course transcriptomes of *B. altitudinis* SCU11 growing in rich medium, as well as that of its progenitor strain BA06 growing in minimal medium, while the latter transcriptomes were sequenced by another group (Supplementary Data 3). The results showed that the number of circular RNA candidates and the abundance of some candidates increased gradually with growth (Fig. 5b, Supplementary Fig. 9b).

Depending on the availability of bacteria, two laboratory strains (*B. subtilis* 168 and *E. coli* K-12 MG1655) and two strains from natural habitats (*B. altitudinis* SCU11 and *B. paralicheniformis*) were used for experimental validation of circular RNA candidates with high abundance. As described above, RNase R-treated total RNA from four strains was used for RT–PCR assays using convergent and divergent primers simultaneously. Finally, 7 out of 9 candidates from *B. altitudinis* SCU11 and 4 out of 5 candidates from *B. paralicheniformis* were identified by divergent primers (Fig. 5c), and RT–PCR product sequencing revealed similar junctions with permuted reads (Supplementary Fig. 10a). However, no candidate was identified by RT-PCR in *B. subtilis* 168 (0/5) or in *E. coli* K-12 MG1655 (0/8). To provide further evidence, five candidates from *B. altitudinis*, including the well-studied SRP RNA, were detected by northern blotting (Fig. 5d, Supplementary Fig. 10b). The expression patterns of the five candidates showed that

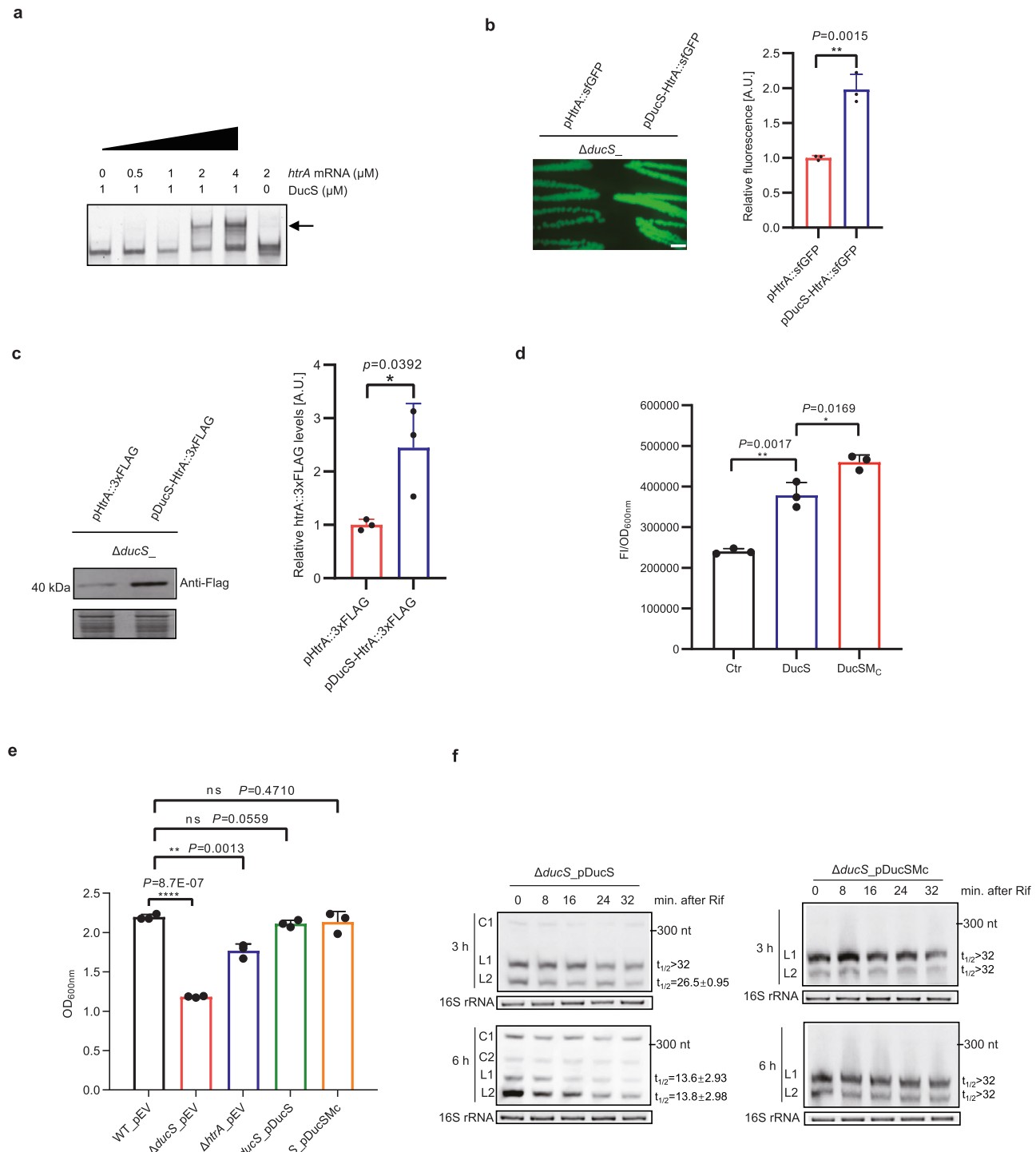

**Fig. 4 | Linear RNAs of DucS promote target *htrA* translation and bacterial oxidative tolerance, while circular RNAs affect the level of linear cognates.**
**a** The DucS-*htrA* complex (arrow) formed in a *htrA* concentration-dependent manner as determined by EMSA ($n = 2$). **b** Colony fluorescence of $\Delta ducS$ strains expressing pHtrA::sfGFP fusion with or without the co-expression of DucS ($n = 3$). Scale bars = 1 mm. Quantitative analysis of fluorescence intensity by ImageJ.
**c** Western blotting revealed that DucS increases HtrA::3×FLAG fusion protein level. Coomassie blue staining of comparable intracellular proteins acted as loading controls (lower) ($n = 3$). Quantitative analysis of band intensity by ImageJ. **d** Effect of circular RNA deficiency (DucSM$_C$) on the regulation of HtrA::sfGFP fusion.

Fluorescence was measured in liquid cultures (8 h) of $\Delta ducS$_pHtrA::sfGFP co-expressing DucS, DucSM$_C$ or not (Ctr). Normalized fluorescence (Fluorescence Intensity/OD$_{600}$) was used for comparison ($n = 3$). **e** Effects of DucS and *htrA* on bacterial growth under H$_2$O$_2$ stress for 4 h ($n = 3$). **f** Stability of DucS linear RNAs was evaluated by rifampicin treatment. EB-stained 16S rRNA was used as a loading control. The half-life ($t_{1/2}$) of L1 and L2 was determined relative to the zero-time point and calculated by pixel counting (ImageJ software). The bars represent the mean ± S.D ($n = 3$). Statistical significance was calculated with a two-tailed unpaired t test. $*P < 0.05$; $**P < 0.01$; $***P < 0.001$; $****P < 0.0001$. Source data are provided as a Source data file.

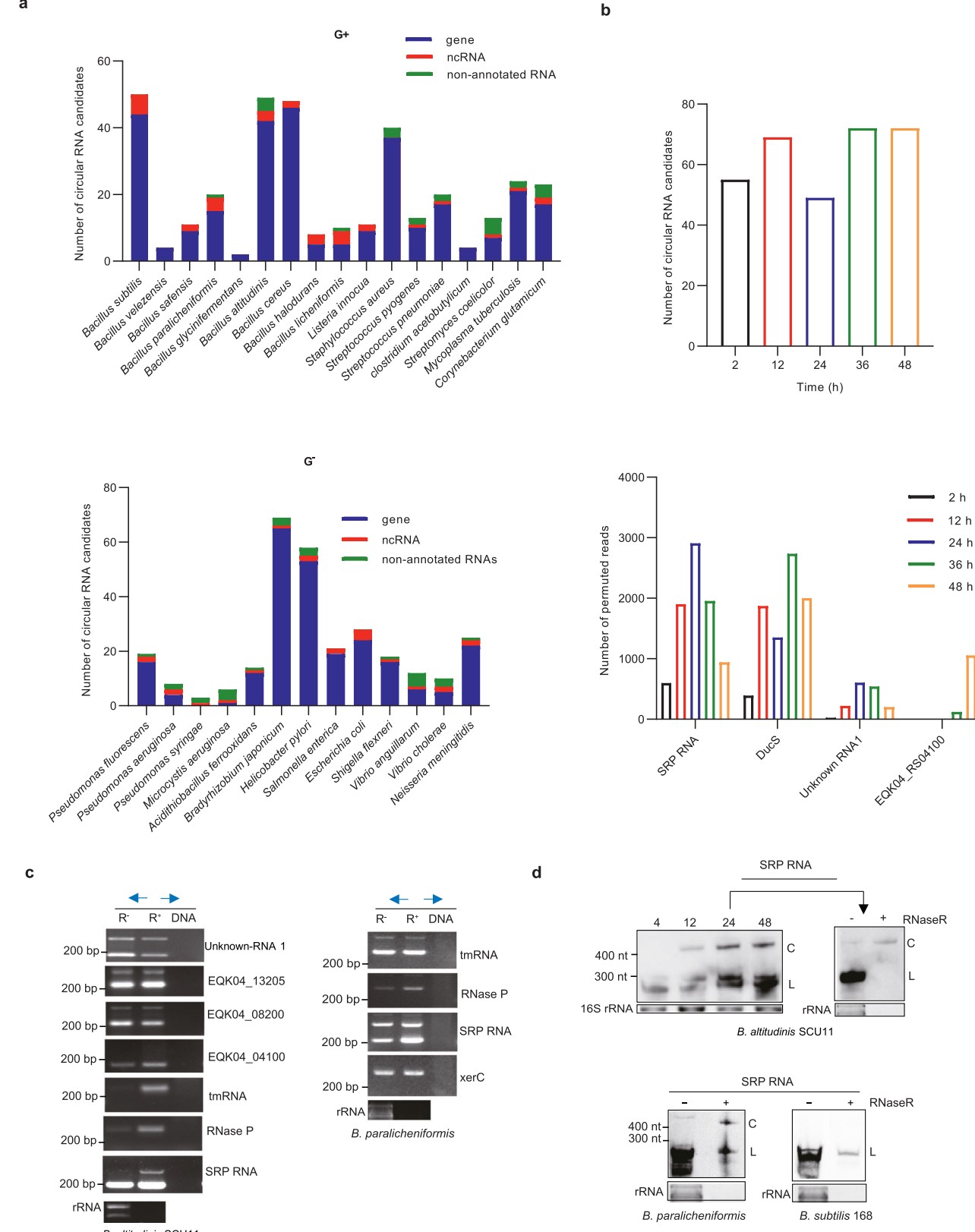

all of them produced extra bands different from the expected linear size, and the extra bands were more resistant to RNase R digestion than linear RNAs, suggesting the existence of circular RNAs. Since the SRP RNA was highly conserved in *Bacillus*, its expression in *B. paralicheniformis* and *B. subtilis* were detected by Northern blotting (Fig. 5d). The results showed that circular SRP RNA existed in

*B. paralicheniformis* but not in *B. subtilis* 168. Moreover, three circular candidates with relatively high abundance were confirmed by junction-spanning probes (Supplementary Fig. 10c). Altogether, these results suggest that more circular RNAs exist in *B. altitudinis* and other bacteria and show growth-related and species-specific expression patterns.

**Fig. 5 | Circular RNAs also exist in other bacteria. a** Numbers of chromosomal loci mapped by more than 5 permuted reads in a chiastic order within a region less than 1000 bp in some species of gram-positive (G⁺) or gram-negative (G⁻) bacteria. **b** Statistics of circular RNA candidate number (upper) and permuted read number of four verified circular RNA loci (lower) in strain *B. altitudinis* SCU11 at different growth phases. **c** Verification of circular RNA candidates by RT–PCR. Blue arrows are divergent primer pairs amplifying circular RNAs. cDNA templates were obtained by reverse transcription of total RNA isolated from *B. altitudinis* SCU11 (8 h, same as that in Fig. 2c) and B. *paralicheniformis* (24 h) pretreated with or without RNase R.

Chromosomal DNA template was used as a negative control. EB-stained rRNA acted as RNase R digestion controls. **d** Verification of circular SRP RNA by northern blotting. Total RNA was extracted from *B. altitudinis* SCU11 (4, 12, 24, 48 h), *B. paralicheniformis* and *B. subtilis* 168 (24 h) cultured in LB medium. 500 ng of total RNA was loaded for northern blotting. Arrows connected RNA samples were treated with RNase R, and EB-stained rRNA acted as RNase R digestion controls. L and C indicate linear and circular RNAs, respectively. Similar results were obtained in two biologically independent experiments (**c, d**). Source data are provided as a Source data file.

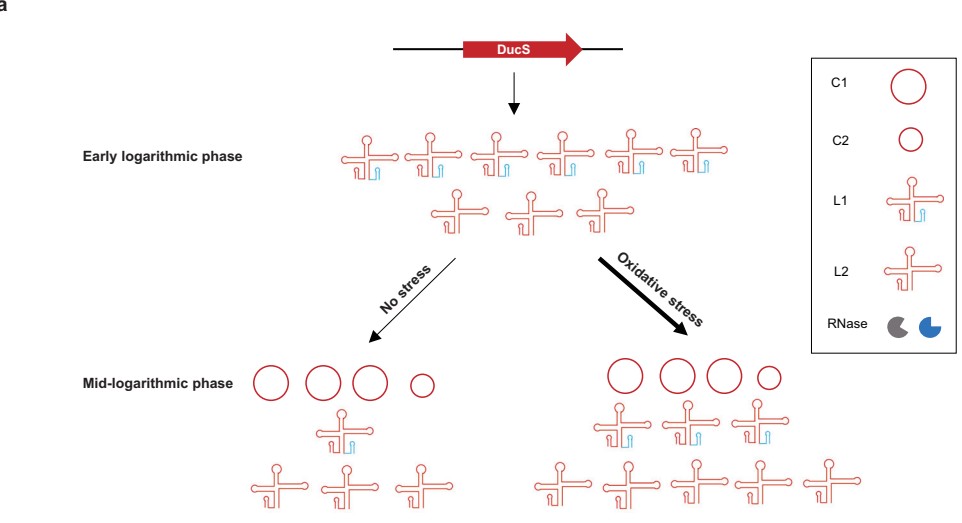

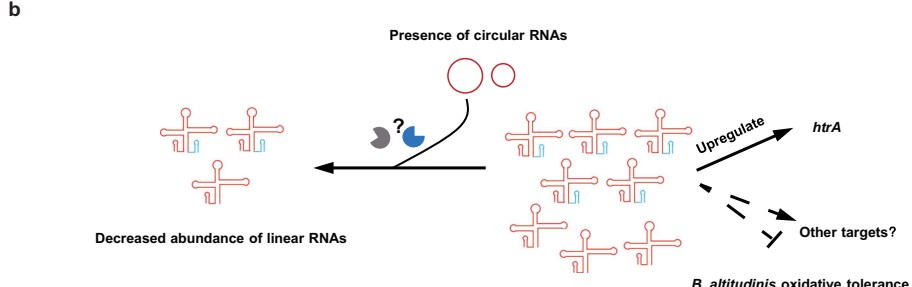

**Fig. 6 | Model of DucS function in *B. altitudinis* SCU11. a** DucS produces different proportions of linear RNAs and circular RNAs depending on bacterial growth phases or stress conditions. Circular RNAs are formed through the circularization of linear RNA L1. Circular RNAs appear at the mid-logarithmic phase and increase gradually thereafter, and H₂O₂ stress can promote the production of linear RNAs.

**b** Circular RNAs and linear RNAs play different roles. The linear RNAs increase bacterial oxidative tolerance by upregulating *htrA* mRNA translation and regulating other targets, while the presence of circular RNAs may decrease the levels of linear RNAs.

## Discussion

Although sRNAs have been characterized as important post-transcriptional regulators in bacteria, all of them are reported to function in linear forms. In this study, we identify a linear and circular dual-conformation regulatory sRNA in *B. altitudinis*, and provide evidence for the existence of other circular RNAs in diverse bacterial species, highlighting the universal existence of circular RNAs in all three domains of life. Our studies demonstrate that DucS produces different proportions of linear and circular RNAs in *B. altitudinis*, depending on bacterial growth phases or stress conditions, and circular RNAs are formed through circularization of linear RNA L1. DucS linear RNAs increase bacterial oxidative tolerance by regulating the *htrA* gene and other targets, while the formation of circular RNAs decreases the level of linear RNAs (Fig. 6). Of course,

further experiments are needed to provide new insights into the regulation and function of DucS.

The biogenesis of nuclear circular RNAs in eukaryotes is tightly controlled by pre-mRNA splicing machinery. However, circular RNA generation in organelles is fairly elusive, even though mitochondrial and chloroplast genome-encoded circular RNAs have been identified and proven to play important roles[40–42]. The intron-driven circularization mechanism may be suitable for some but certainly not all organellar circular RNAs[43], since some show distinct characteristics from this mechanism. Thus, other RNA circularization mechanisms may exist in organelles and need to be explored. Considering that mitochondria and chloroplasts are descendants of endosymbiotic bacteria[44], they may retain some essential prokaryotic features. Although the mechanism of DucS circularization is still unknown, we

ruled out the self-spliced intron pathway and identified critical sequences to circularization.The deletion mutant DucSΔ10-1, which disrupts a predicted stem structure, is unable to undergo circularization. In addition, it is worth noting that DucS C1 and C2 have variable 5'-ends but fixed 3'-ends, and point mutations at $T_{946513}$ do not disrupt the base pairing of the stem but completely interfere with circularization. We speculate that different ribonucleases, such as 5' to 3' exoribonuclease and/or site-specific endoribonuclease, may be involved in DucS circularization. Finally, it is highly likely that bacterial RNA ligases are also involved in RNA circularization. Two classes of RNA ligases, GTP-dependent tRNA-splicing ligase (RtcB)[45] and ATP-dependent RNA ligase (Rnl)[46], have been shown to be involved in RNA circularization in archaea. Direct experimental evidence is required to elucidate circular RNA biogenesis in bacteria, this may also be helpful to reveal the mechanism in eukaryotic organelles.

Advances in high-throughput sequencing and associated bioinformatics tools have facilitated the discovery of circular RNAs, such as studies in eukaryotes, archaea, and bacteria. However, it is well known that the reverse transcription step in RNA-seq library preparation can generate artifactual sequences. Therefore, experimental approaches are required to validate the presence of circular RNAs prior to subsequent functional exploration. In regard to the circular RNA candidates predicted from 30 bacteria in this study, two commonly used methods, divergent RT–PCR and northern blotting in combination with RNase R digestion, are applied for experimental identification. Among the four tested strains, candidates from *E. coli* and *B. subtilis* failed to be detected by RT–PCR. Finally, eleven circular RNA candidates from *B. altitudinis* or *B. paralicheniformis* were found by both methods, and the most interesting candidate was the well-known noncoding SRP RNA, which was detected in both strains. The signal recognition particle (SRP) is a ribonucleoprotein complex conserved in all three domains of life, playing an essential role in the co-translational targeting of secretory and membrane proteins. SRP RNA is pivotal for the structure and function of all SRPs, and the structures of SRP RNA of bacteria, archaea, and eukaryotes show a high degree of similarity to each other[47]. The canonical SRP RNAs exist as linear and structured RNAs whose 5' and 3' regions are clamped together. The circular SRP RNA in *B. altitudinis* shows an increased abundance with growth, even though it is significantly lower than that of the linear forms. Do the circular forms of SRP RNA possess functional potential in bacteria, or are they just transcription noise since they are absent in *B. subtilis*? Considering its increasing tendency with the growth of *B. altitudinis* and the few species of the archaeal genus *Thermoproteus* containing only circular SRP RNAs to function[12], it is worthwhile to explore the contribution of the circular form in the future.

The functional significance of circular RNAs has only been explored preliminarily, even in eukaryotes, which is at least in part due to limitations of the tools used to study them. Similarly, functional characterization of bacterial circular RNAs will be a case-by-case scenario and a huge challenge, and new methods need to be developed for bacteria. Considering that the formation of circular DucS RNAs inevitably leads to a reduction in linear RNA (L1) level, we can speculate that circular RNAs play an indirect regulatory role by antagonizing the level of functional linear RNAs. Given that eukaryotic circular RNAs were initially considered byproducts of RNA splicing and that processing byproducts of bacterial RNAs can function as regulatory sRNAs or RNA sponges[48], it is reasonable to assume that circular RNAs in bacteria, at least those with high abundance, may have yet unidentified biological roles. Several recent reports describe that sRNAs contained in bacterial outer membrane vesicles can be transferred to hosts, such as human airway cells, acting as pathogen-host interkingdom communication tools, as they can modulate host gene expression[49]. Because circular RNAs are very stable and resistant to most RNA-degrading machinery, they may possess special advantages in regulating gene expression intracellularly and extracellularly.

## Methods

### Strains and cultures

The bacterial strains used in this study are listed in Supplementary Data 4. *B. altitudinis* SCU11[21] was obtained by combined mutagenesis from the parent strain BA06 (accession number CGMCCC 0518). This strain was previously classified as *B. pumilus* and recently re-classified to *B. altitudinis* based on genome phylogeny[50]. The plasmids were transferred into *B. altitudinis* by electroporation. Liquid Luria-Bertani (LB, OXOID, England) medium or solid medium containing 1.5% (w/v) agar served as routine cultivation media. Unless indicated otherwise, the liquid cultures were cultivated as follows: the overnight culture was adjusted to $OD_{600nm}$ of 1 and transferred to fresh LB medium at a ratio of 1%, then grown at 37 °C with shaking at $200 \times g$ until indicated period. For the oxidative stress experiments, liquid cultures were grown to $OD_{600nm}$ of 1 and then divided into two aliquots. Equal volume fresh LB was added, and cells were challenged with or without 1.5 mM $H_2O_2$. Where appropriate, antibiotics were supplemented at the following concentrations: 100 μg/ml ampicillin for *E. coli* and 20 μg/ml kanamycin for *B. altitudinis* SCU11.

### Plasmids construction

Plasmids and primers are detailed in Supplementary Data 4 and 5. Plasmid pSU03[51] was used for the stable expression of sRNA or gene in *B. altitudinis*. Plasmid pJOE8999[52] carrying CRISPR-Cas9 system was used for *Bacillus* gene editing. For deletion of the sRNA or gene chromosomal region, DNA fragments upstream and downstream of the target sequence were amplified respectively from *B. altitudinis* SCU11 chromosomal DNA by PCR. Two fragments were fused to a single fragment by overlapping PCR. The fusion fragment with flanking BsaI sites and chemically synthesized sgRNA spacer sequence (designed by http://crispr.dfci.harvard.edu/SSC/) were inserted into plasmid vector pJOE8999. To construct the sRNA expression plasmid, the chromosomal fragment including DucS and its upstream and downstream regions (0–300 bp) was amplified by PCR. The fragments with flanking XbaI/EcoRI sites were cloned into plasmid pSU03. Site-directed mutagenesis of DucS in pSU03 was carried out according to the QuikChange® protocol (Stratagene). The deletion mutations of DucS in pSU03 were generated by amplifying respective parental plasmids using the corresponding divergent primers with homologous sequences. Positive clones were selected on plates with 100 μg/ml ampicillin, and confirmed by sequencing.

The interaction system between DucS and target *htrA* mRNA was constructed as described previously[53]. Briefly, the 5' UTR and the first 27 codons of *htrA* mRNA was amplified from *B. altitudinis* SCU11 chromosomal DNA. The superfolder green fluorescent protein segment (*sfgfp*) was amplified from plasmid pCN33-egfp. Two segments were recombined to form *htrA::sfgfp* fusion by overlapping PCR. The fusion fragment was inserted into Acc651 and EcoRI sites of plasmid pSU03 with or without (control) DucS expression cassette, resulting pHtrA::sfGFP and pDucS-HtrA::sfGFP.

For construction of 3xFLAG-tagged *htrA*, DNA fragments upstream and downstream of the termination codon of *htrA* gene were amplified from *B. altitudinis* SCU11 chromosomal DNA by PCR. The fragments and chemically synthesized 3×FLAG sequence were fused to form *htrA*::3xFLAG by overlapping PCR. The fusion fragment was inserted into the Acc651 and EcoRI sites of pSU03 with or without (control) DucS expression cassette, resulting pHtrA::3xFLAG and pDucS-HtrA::3xFLAG.

### RNA extraction, northern blots and RT–PCR assays

Bacterial cells were harvested by centrifugation at $10,000 \times g$ for 1 min at 4 °C and the pellets were stored at −80 °C. Cell pellets were resuspended in 120 μl of lysozyme (5 mg/ml), and incubated at 37 °C for 10 min. Total RNA was extracted by TRIzol reagent. The purified total RNA was quantified with a NanoDrop ND-2000 spectrophotometer.

For the northern blotting assay, 1–5 µg of total RNA was separated by electrophoresis on an 8% polyacrylamide 1×TBE-urea (7 M) gel. Electrophoresis was conducted at 120 V for 1 h 40 min and then transferred onto a positively charged nylon membrane (GE Healthcare). RNA was cross-linked to the membrane with 1200 mJ of UV. Digoxigenin-labeled RNA probes were prepared by in vitro transcription with a DIG Northern Starter Kit[22] (Roche, Basel, Switzerland). The procedure of hybridization was performed as described previously[54]. Briefly, the membranes were prehybridized for 30 min at 65 °C in DIG Easy Hyb Granules hybridization buffer, then probed with 10 pmol of DIG-labeled probe for 12–16 h. After high-stringency (2×SSC) and low-stringency (0.1×SSC) washes, membranes were blocked and hybridized using blocking solution and Anti-digoxigenin-AP, respectively. Detection was accomplished with a ChemoCam Imager (Intes Science Image Instruments GmbH, Göttingen, Germany). The 16S rRNA was used as the loading control. For RT-PCR assay, the total RNA treated with or without RNase R was reverse-transcribed by random hexamers as primers using the PrimeScript™ RT reagent kit with gDNA Eraser (Takara, Dalian, China). The primers for amplification of selected sequences are listed in Supplementary Data 6. The RT-PCR reactions were performed in 20 µl reaction mixtures including 10 µl Master Mix (Vazyme), 1 µl cDNA (1:50 diluted), 8 µl DEPC-treated water, 0.5 µl of each primer (10 pmol/µl). Afterward, the PCR products were separated on a 2% agarose gel and analyzed by ethidium bromide staining. RT–PCR products from the RNase R-treated sample with divergent primers were cloned and sequenced to detect circularization junction.

### RNase R cleavage assay

RNase R (Epicentre Technologies, USA) was used to degrade linear RNA. Briefly, the total RNA after heating at 95 °C for 5 min and cooling on ice for 2 min was split into two aliquots: one for RNase R digestion and the other for control without digestion[55]. RNase R digestion reaction was carried out at 37 °C for 30 min in 10 µl with 1× reaction buffer [20 mM Tris–HCl (pH 8.0), 0.1 M KCl and 0.1 mM MgCl₂], total RNA (2–10 µg), and RNase R (6–30 U). For control reaction, RNase R was substituted with DEPC-treated water. The reactions were stopped by adding 2 × RNA loading dye (50% glycerol, 7 M urea, 0.01% Xylene green). The treated RNA samples were used directly for northern blotting or purified for RT-PCR.

### Rapid amplification of cDNA ends from circularized RNA (cRACE)

cRACE was used to determine 5′ and 3′ extremities of DucS transcripts by RT-PCR across the ligation site of circularized RNA as template as described previously[23]. Briefly, Total RNA was firstly incubated with Terminator™ 5′-PhosphateDependent Exonuclease (TEX, Epicentre Technologies, USA) at 30 °C for 1 h to remove 5′ monophosphate RNA and heated at 70 °C for 5 min to stop reaction. Then RNA was treated with RNA 5′ pyrophosphohydrolase (RppH, NEB) to convert 5′ triphosphate RNA into 5′ monophosphate RNA and purified by phenol/chloroform extraction. The resulted RNA was then circularized in 50 µl reactions containing 2 µl of T4 RNA ligase I (Takara, Dalian, China), ATP and buffer according to the manufacturer's recommendations. Reactions were allowed to proceed for overnight at 16 °C and purified with phenol/chloroform. The circularized RNA was then subjected to reverse transcription by random primer, PCR amplification with specific primers, PCR products cloning, and sequencing. 5′ and 3′ end fusions were analyzed by mapping DNA sequence of DucS using SnapGene (Version 3.3.1).

### In vitro transcription and electrophoretic mobility shift assay (EMSA)

Interaction regions between DucS and target genes were predicted using online tool IntaRNA (http://rna.informatik.uni-freiburg.de/IntaRNA/Input.jsp). For in vitro transcription assay, template DNA was generated by PCR using gene-specific oligonucleotides with a T7 promoter sequence at the 5′ end of the forward primer. The primers used to amplify DucS sRNA, the 5′ UTR of targets and its nearby sequence are listed in Supplementary Data 5. The PCR fragments were separated by agarose gel electrophoresis and then purified. The In vitro Transcription T7 Kit (Takara, Dalian, China) was used for transcription according to the manufacturer's protocol. Briefly, PCR products with integrated T7 promoters were used as templates. The transcripts were synthesized using 20–200 ng DNA template, 2 µl T7-polymerase buffer (10×), 2 µl ATP (50 mM), 2 µl CTP (50 mM), 2 µl GTP (50 mM), 2 µl UTP (50 mM) and 2 µl T7 RNA Polymerase per 20 µl reaction mixture. Incubated at 42 °C for 1–2 h. The DNA template was removed by 2 µl DNase I treatment at 37 °C for 30 min. An equal volume of phenol (pH 4) was added and centrifuged for 5 min at 13,000 × g, 4 °C. This was then transferred to an equal volume of a 24:1 solution of chloroform/isoamyl alcohol and centrifuged for 5 min at 13,000 × g, 4 °C. RNA was precipitated from the aqueous phase by adding 2 volumes of ethanol and 0.1 volume of 3 M of NaOAc (pH 5.2) solution. RNA precipitation was dissolved in RNase-free water.

An RNA-RNA EMSA assay was performed to verify the interaction of RNAs, as previously described[56,57]. EMSAs were performed with indicated amounts of target mRNAs, DucS RNA, and RNase-free water in 1×REMSA binding buffer (20 mM Tri-HCl [pH 7.3], 20 mM KCl, 1 mM MgCl₂, and 2.4 mM DTT). The reaction mixtures without EMSA binding buffer were incubated for 2 min at 95 °C and placed on ice for 2 min. Then REMSA binding buffer was added and incubated at 37 °C for 30 min. RNA was separated by 8% Native PAGE (0.5× TBE) which was pre-run for 30 min at 100 V and 4 °C. The separation was conducted for 40 min at 100 V and 4 °C, stained with SYBR Gold nucleic acid gel stain (Invitrogen, USA) for 10 min, and visualized under UV light.

### Analysis of self-splicing in vitro

The assays are used to evaluate whether DucS circular RNAs are formed through group II or group I intron processing mechanism. In vitro transcribed RNAs were used for self-splicing assay, performed as previously described[58,59]. Briefly, precursor RNA was heated to 70 °C for 5 min and immediately placed on ice for 3 min, then incubated in 50 mM Tris-HCl (pH 7.5), 10 mM MgCl₂, 10 mM DTT, with (for group I intron) or without (for group II intron) 2 mM GTP at 55 °C (for group I intron) or 45 °C (for group II intron) for 15 min. Then, the products were fractionated on a 4% polyacrylamide gel under denaturing conditions. The separation was conducted for 60 min at 150 V, stained with SYBR Gold nucleic acid gel stain (Invitrogen, USA) for 10 min, and visualized under UV light.

### RNA stability analysis

The overnight cultures of B. altitudinis strains (adjusted to OD₆₀₀nm of 1) were transferred to fresh LB medium at a ratio of 2%, then grown at 37 °C for 3 h and 6 h with shaking at 200 × g. Rifampicin (inhibitor of transcription initiation) was dissolved in dimethylsulfoxide (250 mg/ml) and added to bacterial cultures at a final concentration of 500 µg/ml. Samples were collected after 0, 8, 16, 24, and 32 min and mixed immediately with 0.2 volumes of stop-buffer (95% ethanol, 5% phenol). RNA was extracted for northern blotting analysis. Transcript half-lives ($t_{1/2}$) of DucS L1 and L2 were calculated by pixel counting (ImageJ software) from three biological replicates.

### Western blot analysis

The overnight cultures (adjusted to OD₆₀₀nm of 1) were transferred to fresh LB medium at a ratio of 2%, then grown for 10 h at 37 °C with shaking at 200 × g. Supernatant and cell pellets were harvested from 1 ml culture by centrifugation at 12,000 × g for 10 min at 4 °C. The equal amount of intracellular protein extracted from pellets was separated via SDS-PAGE (12%) gels as loading control. The supernatants representing extracellular proteins were separated via

SDS-PAGE (12%) gels and subsequently transferred onto PVDF membrane (GE Healthcare, UK). The membrane was incubated in blocking solution (2.5 g powdered milk in 50 ml 1× TBS containing 0.1% Tween 20 (TBS-T)) for 1 h. After 2 × washing for 10 min in TBS-T, the membrane was incubated at 4 °C for overnight with the anti-FLAG mAb (ABclonal, China), which was diluted 1:2500 in 1× TBS-T milk. After 2 × washing for 10 min in TBS-T, anti-mouse antibody (1/5000) coupled with Alexa Fluor® 680 Conjugate was incubated at room temperature for 1 h. Azure biosystems C500 was used to detect the signals.

## Fluorescence measurements

*B. altitudinis* strains expressing translational sfGFP-based reporter fusions were grown overnight in LB agar plates for 16–18 h at 37 °C and visualized on Fluorescence stereomicroscope M205FA (Leica, Germany). The overnight cultures (adjusted to $OD_{600nm}$ of 1) of each strain were transferred to fresh LB medium at a ratio of 2%, then grown at 37 °C with shaking at $200 \times g$. To compare expression levels, the fluorescence intensity (FI) and $OD_{600nm}$ of cultures at indicated time after inoculation were measured by microplate reader (BioTek, Synery H1). Control samples not expressing fluorescent proteins were used to subtract background fluorescence. Normalized fluorescence $(FI/OD_{600nm})$[60] was used for comparison. The specific excitation and emission wavelengths for detection of the sfGFP reporter were 448 and 507 nm, respectively.

## DucS permuted reads extraction

RNA-seq BAM file (Accession number SRR8892267) and reference genome (NZ_CP038517.1) of *B. altitudinis* SCU11 were downloaded from NCBI. Based on the sequence of DucS in length of 232 nt, every 30 nt sequence of DucS were aligned to all raw reads, then the high quality matched raw reads (mapped by 30 nt DucS sequence with up to two mismatches) were extracted and mapped to *B. altitudinis* SCU11 genome visualizing by Kblamm software.

## Prediction of circular RNAs in other bacteria

The paired-end RNA-seq BAM files of 30 bacteria were downloaded from NCBI (Supplementary Data 7). Sequencing depths of these RNA-seq samples ranged from 1.1 G bases to 22.8 G bases. The reads were first mapped to corresponding reference genome using STAR[61]. Then, all the unmapped reads were used to identify circular RNAs by CIRCexplorer2[62]. We used a cut-off for circular RNA candidate's statistics when at least five junction reads were detected, and junctions were aligned uniquely to the genome within 1000 nt.

## Statistics

Statistical parameters for the respective experiment are indicated in the corresponding figure legends. Data are presented as the mean ± s.d. calculated using Prism 8.3 (GraphPad Software). $P < 0.05$ was considered statistically significant. Statistical analysis was performed using an unpaired two-tailed Student's t-test with 95% confidence intervals in Prism 8.3. *n* represents the number of biological replicates.

## Computational analysis

30 transcriptomes and genomes of bacteria downloaded from GenBank were used to circular RNA analysis. Targets prediction by CopraRNA in the Supplementary Data 1 is based on the default CopraRNA[33] parameters by extract mRNA regions of 300 nt (from 200 nt upstream to 100 nt downstream of the start codon), from 23 homologous strains searched by GLASSgo[34]. Interactions between targets and DucS were predicted by IntaRNA[63]. Promoter and terminator region were predicted by BPROM (http://www.softberry.com/berry.phtml?topic=bprom&group=programs&subgroup=gfindb) or FindTerm (http://www.softberry.com/berry.phtml?topic=findterm&group=programs&subgroup=gfindb).

## Reporting summary

Further information on research design is available in the Nature Portfolio Reporting Summary linked to this article.

## Data availability

The raw RNA-seq data of other strains used in this study are available in the NCBI with the accession codes listed in Supplementary Data 7. Data supporting the findings of this work are available within the paper and its Supplementary Information files. Source data for this paper have been submitted to Figshare [https://doi.org/10.6084/m9.figshare.23660757][64], which includes all uncropped and unprocessed scans. Source data are provided with this paper.

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

## Acknowledgements

We thank Y. Liu, F. Huang, and CB. Zhang for providing us with *B. paralicheniformis*, *B. subtilis* 168, and *E. coli* MG1655 strains, respectively. We also thank Professor S. Chabelskaya for providing us plasmid with sfGFP. We thank Life Science Core Facilities (College of Life Sciences, Sichuan University) for providing us equipments and H. Kang for technical support. This work was funded by Sichuan Science and Technology Program (2019YFG0273) and the National Natural Science Foundation of China (32060720). We thank Y. Zhao for comments on the manuscript.

## Author contributions

T.-T.H. supervised the project, contributed to the design of the experiments, conducted most of the experiments and analysis and wrote the manuscript. Y.-F.X. designed and conducted circular DucS verification experiments and their analysis and wrote the manuscript. X.L. performed RNA extracted and the northern blot analysis shown in Supplementary Fig. 10. X.W., H.-S.C. and D.O.-Y. performed transcriptome data analysis. J.-Y.L. conducted gene htrA knockout and EMSA assay shown in Supplementary Fig. 5. H.-Y.L. performed RNA extracted and northern blot analysis shown in Fig. 5d. J.Q. and Y.H. contributed to the initial phase of this project. H.-Y.W. supervised the project, contributed to the design of the experiments, analyzed the data, and wrote the manuscript.

## Competing interests

The authors declare no competing interests.
