## [Peer Review File · Nature Communications]

A linear and circular dual-conformation noncoding RNA involved in oxidative stress tolerance in *Bacillus altitudinis*Reviewer #1 (Remarks to the Author):

NCOMMS-23-08914

A linear and circular dual-conformation noncoding RNA involves in oxidative stress tolerance of *Bacillus altitudinis*. He et al. and Wang

The authors describe an RNA termed DucS that exists in linear and circular forms. Some sequences in DucS RNA are identified that influence circularization efficiency, but no potential mechanisms or structural information is provided. He et al. present some evidence that DucS might be involved in tolerance to oxidative stress, and that the circular isoforms might participate in targeting the linear versions for degradation. The authors conclude with some specific examples suggesting that circular RNAs might generally be more pervasive among bacteria.

The authors use bioinformatic and experimental methods to establish that a subset of DucS RNAs exist as circles. But no structural framework is presented that would help explain why this is the case. Although it is not mentioned by the authors, inspection of the DucS sequence reveals regions of extensive complementarity near the 5' and 3' ends of the RNA, which would result in anchoring the DucS RNA termini in proximity to one another. Specifically, nucleotides 946338-946346 are perfectly complementary to nucleotides 946482-946490 (red highlighting in attached figure) and nucleotides 946347-946356 are complementary to nucleotides 946499-946508 (blue highlighting in attached figure). This results in the formation of 19 base pairs, likely creating a stable RNA structure in which the 5' and 3' ends are held near one another (see secondary structure schematic also in the attached figure).

This predicted RNA structure is almost certain to be the substrate for whatever ligase activity is responsible for generating the circular isoforms. Indeed, the authors' own observations suggest this is the case. For example, many of the junction sites occur in the S2 region (Figure 2d and Supplementary Figure 4b), which corresponds exactly to nucleotides 946338-946346. In addition, DucS Δ 10-1, in which most of the sequence corresponding to B' in the attached figure is deleted, is no longer able to undergo circularization (Figure 3e).

The critical question, then, is whether this RNA structure is merely a fortuitous occurrence, and just happens to serve as a substrate for an unknown ligase activity, or whether it is biologically pertinent, having been selected for its ability to promote circularization. The authors suggest a biological role in oxidative stress response, but the evidence for this is weak.

He et al. state that DucS is narrowly conserved in just a few related species of what is known as the *Bacillus pumilus* group. But these species, which include *B. altitudinis*, *B. safensis*, *B. xiamenensis*, and *B. cellulasensis*, are highly similar to one another, and their DucS sequences, shown in Supplementary Figure 1, are essentially identical. Because this sequence is not found in any other bacteria, it would be just as accurate to state that DucS is not conserved at all.

The authors computationally identify htrA as a potential target of DucS and provide some evidence that DucS induces modest levels of htrA gene expression (Figure 4). However, even if this evidence is to be believed, He et al. show that the linear forms of DucS suffice to mediate these effects, leaving the circular versions of DucS RNA with no specific biological function. Also in Figure 4, the authors show that the half-lives of linear DucS RNAs are slightly shorter in the presence of DucS circles than in their absence, and then posit a regulatory role for circular DucS RNAs involving decay of the linear versions. But it is not at all clear what biological function this would serve. He et al. never show that circular DucS is itself an ineffective regulator of htrA expression. Their data simply suggests that exclusively linear DucS might induce slightly more htrA expression than a mixture of circular and linear forms (Figure 4d).

The authors conclude with data from a bioinformatic search for additional circular RNAs in bacteria, and also provide experimental evidence for the existence of some of these RNA circles (Figure 5). It is worth noting that many of these RNAs, including RNase P, SRP RNA, and tmRNA, have secondary structures placing their 5' and 3' termini in close proximity. It is most likely this feature, which is shared by DucS, that drives the circularization of such RNAs. Thus, the data in Figure 5 would suggest that many of the instances of circular RNAs in bacteria are simply artifacts resulting

from the proximity of their termini. This erodes the already weak claim that circular DucS has a defined biological role.

Reviewer #1 Attachment on the following page

Graphics accompanying review of NCOMMS-23-08914: “A linear and circular dual-conformation noncoding RNA involves in oxidative stress tolerance of *Bacillus altitudinis*”. (He et al. and Wang.)

DucS

```

946261 tatctttaa aaaagggttg tcgaagctcg taaattctgc gataatactt attgaaagtg
946321 ctttcaaaaa gcgctatcgt gatgaattgg gtgcgttcca ctttggattt acggcgatga
946381 gccaaatgag agagaagcgc ttctgggaga aaagcaaagc ataaggggat atgatataag
946441 caatgctttg gtgtgtgaaa gatgagggga tcatggggag cttcatcacc ccattgttac
946501 gcatccaaac caacataaaa aaaaagccta ggggtaggct tttttattt gcgattgagg
946561 cataaaacaa gggctgcttc aactgcatgt tctaaagcga taagctaggg catatcctca
946621 tcgtaagtac agtcatcaag aagggggatg cctcag
  
```

Regions highlighted in red are complementary to one another, as are the regions highlighted in blue. Selected 5' and 3' termini that were identified by the authors are highlighted in green.

A secondary structure model of DucS RNA (below), which results in the 5' and 3' ends being anchored in proximity to one another.

DucS RNA structure based on the complementarity shown above:

Reviewer #2 (Remarks to the Author):

With DucS from *Bacillus altitudinis*, the authors discovered the first example of a bacterial noncoding sRNA which can be present in either a linear (L1 and L2) or circular (C1 and C2) conformation. The ratios of all four conformations differ over growth and under stress conditions. The authors show that the nucleotide at the DucS 5' end must be a T to allow circularization, and that different internal regions of the sRNA impact the amount of circular DucS. They also mapped the 3' ends which are decisive for the two linear and the two circular forms of the sRNA. Furthermore, they employed an assay used for the investigation of group I and group II self-splicing introns and found no indication for self-splicing. Not unexpectedly, only the linear forms can act as regulatory sRNAs on at least three target mRNAs, of which *htrA* mRNA encoding a serine protease was investigated in more detail. Linear DucS binds far upstream of the *htrA* RBS and activates translation by an unknown mechanism, but does not affect the amount of *htrA* mRNA. As *htrA* was known to be involved in stress response, the authors compared the growth of DucS wild-type, knockout and complementation strains under different stress conditions and suggest that DucS could promote hydrogen peroxide tolerance in *B. altitudinis* via its linear form.

The authors propose that the generation of circular forms does not simply reduce the amount of linear forms, but that circular forms promote the degradation of the linear forms, which is supported by half-life measurements in circularization deficient mutants.

They searched for potentially circular sRNAs in transcriptome data of 30 species and found a number of them in both Gram-positive and Gram-negative bacteria, interestingly, among them in *B. licheniformis* and *B. paralicheniformis*, but not in the model organisms *B. subtilis* 168 or in *E. coli*.

The authors' discovery and characterization of the first circular noncoding RNAs in bacteria is exciting and of great interest for the reader of *Nature Communication*. The paper is clearly written (although the English needs to be improved by a professional language service), and all findings are supported by figures in the main text and in the Supplementary data.

However, there are a few points that I detail below which should be clarified before a revised version of the manuscript can be accepted.

Major points

1) Which enzymes are involved in the circularization of DucS? Since the authors excluded self-splicing, I assume at least a ligase must be involved, and most probably an RNase. *B. subtilis* has an essential NAD-dependent ligase, LigA, and two ATP-dependent ligases, among them LigB. I propose to take a look at the genome of *B. altitudinis* and search for similar genes. Furthermore, all firmicutes have the same set of main ribonucleases, namely RNase Y as the main endoribonuclease, RNases J1 and J2 as endo/5'-3' exoribonucleases, PNPase as the main 3'-5' exoribonuclease and RNase R as another 3'-5' exoribonuclease. I suggest to construct CRISPR-Cas knockdown strains for Y, J1 and PnpA and perform Northern blots with these strains to find out if in one of these strains the ratio of the 4 DucS species differs from that in the isogenic wild-type strain.

2) Interestingly, the authors did not observe any circular sRNA in *B. subtilis* 168. I propose they express DucS from a plasmid in *B. subtilis* and analyze in Northern blots if this sRNA is also present in a circular form or only in a linear form. The latter would indicate that enzymes for circularization are different in both species and, at the same time, corroborate that self-splicing can be excluded. However, if DucS is also present in a circular conformation, it would make the search for responsible enzymes much easier, because *ligA* and *ligB* mutants as well as RNase knockout or knockdown strains are available for *B. subtilis*.

3) The authors observe that the percentage of circular DucS increases over growth. However, what is the signal/trigger for circularization? Most probably, the signal is somehow linked to the expression of an enzyme/protein involved in circularization.

4) How can the circular conformation of the same sRNA promote degradation of the linear

conformation? Do the authors have any hypothesis or possible explanation?

Minor comments:

1) Just a question: The authors used denaturing PAA gels and detected that circular DucS run slower than linear. Did they also take a look in agarose gels? There, circular RNA runs much faster than linear one.

2) Not only a, b, c, d and e used for labelling the Figures, but also the letter size used for labelling within the figures has to be increased. For example, the subscripted letters in Fig. 3 b, c, d and e are barely decipherable. A possible solution to this problem would be to use no subscript, but simply DucS Δ 16 etc.

3) line 77 reference 21 This reference is about *Bacillus pumilus*, not *B. altitudinis*

4) line 110 reference 22. Is this the correct reference?

5) line 211 The references 32,33 are not only for CopraRNA (this is ref. 32), but also for another program, GLASSgo (ref. 33), please correct

5) The English needs to be corrected by a professional language service. I provide a few examples:

Manuscript title: It must read: "involved in" or "is involved in" and not "involves"

line 92 must be corrected into: A DucS knockout strain..was constructed through the CRISPR-Cas9 system and the complementation strain was.....plasmidborne DucS into the Δ ducS strain

line 116 not "slowly shifting transcripts", but "slowly migrating transcripts"

line 116/117 not "made us confusing" but "confused us"

line 131 Correct: " In view of band1 and band2 appeared in 8 h sample" into "Concerning band 1 and band 2 that appeared in the 8 h sample...."

Reviewer #3 (Remarks to the Author):

This manuscript characterizes a new functional circular RNA in *Bacillus altitudinis* and indicates that a number of other circular RNAs exist in bacteria. The experiments are clear, well controlled, and convincing. The work should spawn others to look for and characterize bacterial circular RNAs and thus the manuscript represents a critical starting point for the field. I have several suggestions for clarifying the results.

(1) There are a number of places where important control data are not shown. The data on Line 134 should be shown to confirm the authors' interpretation regarding the 3' end, the data in Line 165 need to be shown to prove that RNase R is behaving as expected, the data in Line 223 should be shown to prove that DucS does not activate all transgenes, and the RT-PCR sequencing data should be shown in Line 304.

(2) Fig 4f: RNA half lives should be more accurately determined rather than simply writing >32 as has been done in many cases.

(3) The manuscript should be thoroughly checked for grammatical errors. For example, "involves" in the title (Line 2) should be "involved".

(4) A supplemental table with the sequences of all circRNA candidates and the number of detected sequencing reads for each (Fig 5) should be provided.

Minor points:

(1) Line 62: Circular RNAs in eukaryotes were discovered more than 30 years ago, e.g. Nigro et al

(1991) Cell "Scrambled exons" and Capel et al (1993) Cell "Circular transcripts of the testis-determining gene Sry in adult mouse testis".

(2) Figures 4b, 4c, S7 should be quantified.

(3) Fig 5d, Supp Fig 10: It would be good to also use a Northern probe that spans the junction as one additional way of proving that at least some of the identified transcripts are circular RNAs.

(4) Line 354: It is known that RNase R does not digest all linear RNAs (PMID: 28444238, 31269210) so this may be the reason why some transcripts appear circular by RNase R Northern but not by RT-PCR.

A linear and circular dual-conformation noncoding RNA involved in the oxidative stress tolerance of *Bacillus altitudinis*

(NCOMMS-23-08914)

Response to the reviewers' comments

Dear editors and reviewers,

We would like to thank you for your letter and suggestions, which are very helpful for the improvement of our manuscript. All reviewers' comments have been taken into account and we attempted to answer all questions. We have made corresponding revisions in our manuscript, showing the main changes in the red text. We have copied and pasted all referees' comments below, and addressed each one individually. The original comments are in italic with black color, and the responses are in normal font with blue color. As you will see, we have made every attempt to incorporate these suggestions as thoroughly as possible.

Point-by-point responses to the Reviewers' comments

Reviewer #1:

Comments: *The authors describe an RNA termed DucS that exists in linear and circular forms. Some sequences in DucS RNA are identified that influence circularization efficiency, but no potential mechanisms or structural information is provided. He et al. present some evidence that DucS might be involved in tolerance to oxidative stress, and that the circular isoforms might participate in targeting the linear versions for degradation. The authors conclude with some specific examples suggesting that circular RNAs might generally be more pervasive among bacteria.*

The authors use bioinformatic and experimental methods to establish that a subset of DucS RNAs exist as circles. But no structural framework is presented that would help explain why this is the case. Although it is not mentioned by the authors, inspection of the DucS sequence reveals regions of extensive complementarity near the 5' and 3' ends of the RNA, which would result in anchoring the DucS RNA termini in proximity to one another. Specifically, nucleotides 946338-946346 are perfectly complementary to nucleotides 946482-946490 (red highlighting in attached figure) and nucleotides 946347-946356 are complementary to nucleotides 946499-946508 (blue highlighting in attached figure). This results in the formation of 19 base pairs, likely creating a stable RNA structure in which the 5' and 3' ends are held near one another (see secondary structure schematic also in the attached figure).

This predicted RNA structure is almost certain to be the substrate for whatever ligase activity is responsible for generating the circular isoforms. Indeed, the authors' own observations suggest this is the case. For example, many of the junction sites occur in the S2 region (Figure 2d and Supplementary Figure 4b), which corresponds exactly to nucleotides 946338-946346. In addition, DucS Δ 10-1, in which most of the sequence corresponding to B' in the attached figure is deleted, is no longer able to undergo circularization (Figure 3e).

The critical question, then, is whether this RNA structure is merely a fortuitous occurrence, and just happens to serve as a substrate for an unknown ligase activity, or whether it is biologically pertinent, having been selected for its ability to promote circularization. The authors suggest a biological role in oxidative stress response, but the evidence for this is weak.

He et al. state that DucS is narrowly conserved in just a few related species of what is known

as the *Bacillus pumilus* group. But these species, which include *B. altitudinis*, *B. safensis*, *B. xiamenensis*, and *B. cellulasensis*, are highly similar to one another, and their *DucS* sequences, shown in Supplementary Figure 1, are essentially identical. Because this sequence is not found in any other bacteria, it would be just as accurate to state that *DucS* is not conserved at all. The authors computationally identify *htrA* as a potential target of *DucS* and provide some evidence that *DucS* induces modest levels of *htrA* gene expression (Figure 4). However, even if this evidence is to be believed, He et al. show that the linear forms of *DucS* suffice to mediate these effects, leaving the circular versions of *DucS* RNA with no specific biological function. Also in Figure 4, the authors show that the half-lives of linear *DucS* RNAs are slightly shorter in the presence of *DucS* circles than in their absence, and then posit a regulatory role for circular *DucS* RNAs involving decay of the linear versions. But it is not at all clear what biological function this would serve. He et al. never show that circular *DucS* is itself an ineffective regulator of *htrA* expression. Their data simply suggests that exclusively linear *DucS* might induce slightly more *htrA* expression than a mixture of circular and linear forms (Figure 4d).

The authors conclude with data from a bioinformatic search for additional circular RNAs in bacteria, and also provide experimental evidence for the existence of some of these RNA circles (Figure 5). It is worth noting that many of these RNAs, including RNase P, SRP RNA, and tmRNA, have secondary structures placing their 5' and 3' termini in close proximity. It is most likely this feature, which is shared by *DucS*, that drives the circularization of such RNAs. Thus, the data in Figure 5 would suggest that many of the instances of circular RNAs in bacteria are simply artifacts resulting from the proximity of their termini. This erodes the already weak claim that circular *DucS* has a defined biological role.

Response: We do appreciate the reviewer's comments and suggestions on this manuscript, especially professional RNA structural analysis. Since the comments of reviewer #1 are not made one by one, we will respond mainly from two aspects, structure and function of circular RNAs.

In terms of RNA structure, as well as the relevant circularization mechanism, we fully agree with the comment that RNA structure is critical to its stability and circularization. Due to the limitation of figure size, the predicted secondary structure of *DucS* was not provided in the previous version of the manuscript. According to the reviewer's suggestion, we also predict the

secondary structure of other experimentally validated circular RNAs by two common tools, RNAfold and Mfold. In general, different programs give rise to similar but not identical base-pairing pattern. The structure comparison is focus on DucS and SRP RNA of *B. altitudinis*, since their 5' and 3' terminal sequences have been determined by cRACE, rather than just predictions made from transcriptome data. Based on the structure comparison (Extended Data Fig. 1) and other experimental evidence, we can infer the probable circularization mechanism, at least for DucS and SRP RNA, in *B. altitudinis*. Just as the reviewer indicated, the close proximity of RNA termini contributes to RNA circularization. Since the potential circularization mechanism is helpful to understand circular RNAs in bacteria, we have supplemented the relevant analysis and figure in **Discussion** of the revised manuscript (Line 346-372, Supplementary Fig. 11), as follows on:

“Comparing the predicted structures of DucS and SRP RNA by RNAFold, one remarkable structural element is the terminal stem, where the circularization junctions located (Supplementary Fig. 11). It was revealed that maturation of SRP RNA in *B. subtilis* was accomplished by cleavage of several nucleases, including RNase III, at the terminal stem of precursor RNA. Considering the high conservation of SRP RNA in *Bacillus*, as well as the fact that circularization junctions of SRP RNA in *B. altitudinis* and *B. paralicheniformis* match exactly to the termini of mature linear SRP RNA of *B. subtilis*, we suggest that circularization of SRP RNA is probably mediated by base-pairing of RNA termini, and following nucleases processing. It is plausible that the close proximity of RNA termini would contribute to RNA circularization. In eukaryote, circRNAs are generated through back-splicing by the canonical spliceosomal machinery which is facilitated by RNA pairing or RNA-binding proteins (RBP). In archaea, circularization of SRP RNA and box C/D sRNA require the base-pairing of RNA termini. So far as DucS, in addition to the similar terminal stem, it is worth noting that circular C1 and C2 RNAs have variable 5'-ends but fixed 3'-ends, and point mutations at T₉₄₆₅₁₃ don't disrupt base-pairing of the stem but completely interfere with circularization. We speculate that different ribonucleases, such as 5' to 3' exoribonuclease, and/or site-specific endoribonuclease, may be involved in DucS circularization. It looks like circularization through base-pairing and ribonucleases processing is a simple and effective pathway in bacteria, but there should exist other mechanisms, for example, through RBPs to tether RNA termini into close proximity since terminal stems are not found in all candidates. In addition, it is highly likely that bacterial RNA ligases are also involved in RNA circularization. Two classes of RNA ligases, GTP-dependent tRNA-splicing ligases (RtcB) and ATP-dependent RNA ligase (Rnl), have been shown to be involved in RNA circularization in archaea. Direct experimental evidence is required to elucidate circular RNA biogenesis in bacteria; this may also be helpful to reveal the mechanism in eukaryotic organelles.”

Extended Data Fig. 1 Secondary structures of *B. altitudinis* DucS and other identified candidates predicted by RNAFold. Blue and red circles indicate 5' and 3' circularization junctions identified by RT-PCR, respectively.

We also agree that circular RNAs revealed by a computational approach may result from artifacts. However, we don't think that the experimentally validated circular RNAs in bacteria come only from fortuitous events. RNA circularization may occur accidentally, but in some cases may also be biologically pertinent, although not much is known right now.

In terms of the function of circular DucS, up to now, we have not yet solid evidences to support its biological role. Even though we observed a correlation between the occurrence of circular DucS and the decreased half-lives of linear DucS, we couldn't conclude that circular DucS involving decay of the linear versions, since short of directed evidence to support the potential mechanism. Therefore, we have deleted the overstretched conclusion from the **Abstract** and throughout the paper, and replaced with describing the phenomenon only and providing some interpretations in the **Discussion** of the revised manuscript (Line 402-404, 408-409).

However, there are several points to help us understanding the potential biological functions of circular DucS as well as other candidates in bacteria.

1) In case of DucS, we have indicated in the manuscript that circular RNAs are converted only from L1 since L2 is absent from the 3' junction site; Thus, the formation of circular RNAs in wildtype inevitably leads to a reduction in L1 level. Then, we can speculate that circular RNA plays an undirected role though antagonizing the level of regulatory linear RNA.

2) In view of the observations that the occurrence of circular DucS and SRP RNA follow a constant emerging pattern (Figs. 1b and 5d), even in different species (for SRP RNA), we speculate that it is biological rather than fortuitous events that promote circularization.

3) Given that eukaryotic circular RNAs are initially considered as byproducts of RNA splicing, it is reasonable to assume that circular RNAs in bacteria, at least those with high abundance and constant pattern, may have yet unidentified biological roles.

4) The most obvious advantage of RNA being in a circular form rather than traditional linear one is stability, being protected against the degradation of ubiquitously spread exoribonucleases. Perhaps a function of circularization is to secure the termini and essential secondary structure of RNA under certain special circumstances.

5) Circular RNAs play roles in eukaryotes and archaea. From the point of view on evolution, the circular form may be not only favorable for eukaryotes and archaea, but also should be

beneficial to bacteria.

6) The significance of this work is the verification of the existence of circular RNAs in bacteria.

We hope these finds will encourage others to this field, to look for and characterize bacterial circular RNAs, to develop new approaches for in-depth analysis of these challenging RNAs. As the field moves forward, the function of circular RNAs in bacteria will be eventually revealed in the future.

In terms of reviewer's comment on Figure 4d, at this point, we don't have directed evidence to support that circular DucS is itself an ineffective regulator of *htrA* expression. We agree that this is a potential limitation of this study. We hope, in the future, to employ more experiments to answer this question, for example, through EMSA to compare the binding of *htrA* mRNA with linear or circular DucS.

Finally, we agree with the reviewer's assessment that we used an inaccurate description as to the conservation of DucS (Line 87-89). Accordingly, we rephrased the sentence like this: "DucS is encoded by the intergenic region of *B. altitudinis* chromosome that exists only in several closely related species, including *B. pumilus*, *B. cellulasensis*, *B. safensis* and *B. xiamenensis*."

Reviewer #2

Comments: *With DucS from Bacillus altitudinis, the authors discovered the first example of a bacterial noncoding sRNA which can be present in either a linear (L1 and L2) or circular (C1 and C2) conformation. The ratios of all four conformations differ over growth and under stress conditions. The authors show that the nucleotide at the DucS 5' end must be a T to allow circularization, and that different internal regions of the sRNA impact the amount of circular DucS. They also mapped the 3' ends which are decisive for the two linear and the two circular forms of the sRNA. Furthermore, they employed an assay used for the investigation of group I and group II self-splicing introns and found no indication for selfsplicing. Not unexpectedly, only the linear forms can act as regulatory sRNAs on at least three target mRNAs, of which htrA mRNA encoding a serine protease was investigated in more detail. Linear DucS binds far upstream of the htrA RBS and activates translation by an unknown mechanism, but does not affect the amount of htrA mRNA. As htrA was known to be involved in stress response, the authors compared the growth of DucS wild-type, knockout and complementation strains under different stress conditions and suggest that DucS could promote hydrogen peroxide tolerance in B. altitudinis via its linear form.*

The authors propose that the generation of circular forms does not simply reduce the amount of linear forms, but that circular forms promote the degradation of the linear forms, which is supported by half-life measurements in circularization deficient mutants.

They searched for potentially circular sRNAs in transcriptome data of 30 species and found a number of them in both Gram-positive and Gramnegative bacteria, interestingly, among them in B. licheniformis and B. paralicheniformis, but not in the model organisms B. subtilis 168 or in E. coli.

The authors' discovery and characterization of the first circular noncoding RNAs in bacteria is exciting and of great interest for the reader of Nature Communication. The paper is clearly written (although the English needs to be improved by a professional language service), and all findings are supported by figures in the main text and in the Supplementary data.

However, there are a few points that I detail below which should be clarified before a revised version of the manuscript can be accepted.

Major points:

1. Which enzymes are involved in the circularization of DucS? Since the authors excluded self-splicing, I assume at least a ligase must be involved, and most probably an RNase. B. subtilis has an essential NAD-dependent ligase, LigA, and two ATP-dependent ligases, among them LigB. I propose to take a look at the genome of B. altitudinis and search for similar genes. Furthermore, all firmicutes have the same set of main ribonucleases, namely RNase Y as the main endoribonuclease, RNases J1 and J2 as endo/5'-3' exoribonucleases, PNPase as the main 3'-5' exoribonuclease and RNase R as another 3'-5' exoribonuclease. I suggest to construct CRISPR-Cas knockdown strains for Y, J1 and PnpA and perform Northern blots with these strains to find out if in one of these strains the ratio of the 4 DucS species differs from that in the isogenic wild-type strain.

Response: Thank you very much for your professional suggestions. We quite agree with your opinion that a ligase may be involved in circularization of RNA. The classical ATP-dependent RNA ligases are classified in six different families (Rnl1-Rnl6), present respectively in different organisms ^[1]. In archaea, two unrelated families of RNA ligases have been identified to be involved in RNA circularization in different species. RtcB proteins, structurally unrelated to classical ATP-dependent RNA ligases, function in most archaea as GTP-dependent tRNA-splicing ligases to form mature-sized tRNAs, as well as circular tRNA-introns ^[2]. At least in some archaeal species, ATP-dependent RNA ligases from Rnl3 family are responsible for box C/D RNA circularization and rRNA processing ^[3]. Therefore, it is highly likely that bacterial RNA ligases may also involve in the RNA circularization. The best case for a *bona fide* bacterial RNA repair pathway, identified in several bacterial species, is Pnkp–Hen1 complex, in which Pnkp (LQRI_RS01820) is a multifunctional enzyme composed of three catalytic domains: N-terminal kinase, central phosphoesterase, and C-terminal RNA ligase (Rnl4 family) ^[4]. However, the Pnkp–Hen1 cassette is absent in the majority of bacterial species. *Escherichia coli*, as well as *Salmonella Typhimurium*, exploit RtcAB complex to modify RNA ends (RtcA) and carry out ligation (RtcB) ^[5]. Owing to the incomplete genome annotation, no RNA ligase was annotated in *B. altitudinis* SCU11. In *B. subtilis* 168, protein YjcG was annotated as putative RNA ligase. Then its homolog (WP_017359656.1) in *B. altitudinis* was knocked out, but the circularization of DucS was

not affected (Extended Data Fig. 1a). Through Protein Blast, no homolog of RtcB was detected, and two homologs of Pnkp protein were found in *B. altitudinis*, PrpE (WP_017367020.1) and metallophosphoesterase (WP_017367319.1). The PrpE contains only phosphoesterase domain and can be excluded. The metallophosphoesterase contains three catalytic domains as Pnkp while the RNA ligase is from Rnl1 family, and whether it is related to RNA circularization needs further experimental verification. In addition, since modern RNA and DNA ligases are thought evolved from an ancestral ATP-using nucleotidyltransferase (NTase) domain [6], we are uncertain if DNA ligases, ligA and ligD, involved in RNA ligation in *Bacillus*.

We also quite agree with reviewer's idea that ribonucleases (RNases) may be involved in circular RNA formation. There are over 20 RNases have been revealed in *B. subtilis*, and only nine are shared by *E. coli* [7]. RNase III (encoded by *rnc* gene), a double-strand-specific endoribonuclease, was firstly selected for gene knockout to test its effect on RNA circularization. Data showed that deficiency of RNase III could moderately decrease the abundance of circular RNA (Extended data Fig. 1b). We agree that more ribonuclease-disruptions would be helpful to understand details of circularization, especially on those main ribonucleases as the reviewer suggested, and the recently reported Mini-III, a dsRNA endoribonuclease showing a degree of sequence specificity [8]. Considering the probable functional redundancy of different ligases and ribonucleases, double or triple knockouts may be needed to reveal the roles of these enzymes on RNA circularization in bacteria. Work on this part is going on and may be time-consuming.

Extended Data Fig. 1 Northern blotting to analyze expression abundance of DucS transcripts in *B. altitudinis* SCU11 (wildtype, WT) and its derivatives $\Delta yjcG$, Δrnc and Δrnc carrying a plasmid expressing *rnc* (*pnrc*). Total RNA was extracted from each strain from culture at indicated time points.

Reference:

- [1] Becker H.F, L'Hermitte-Stead C, Mylykallio H. Diversity of circular RNAs and RNA ligases in archaeal cells. *Biochimie*. **164**, 37-44 (2019).
- [2] Englert M, Sheppard K, Aslanian A, Yates J.R 3rd, Söll D. Archaeal 3'-phosphate RNA splicing ligase characterization identifies the missing component in tRNA maturation. *Proc. Natl. Acad. Sci. USA*. **108**, 1290-5 (2011).
- [3] Becker H.F, Héliou A, Djaout K, Lestini R, Regnier M, Mylykallio H. High-throughput sequencing reveals circular substrates for an archaeal RNA ligase. *RNA Biol*. **14**, 1075-1085 (2017).
- [4] Wang P, Chan C.M, Christensen D, Zhang C, Selvadurai K, Huang R.H. Molecular basis of bacterial protein Hen1 activating the ligase activity of bacterial protein Pnkp for RNA repair. *Proc. Natl. Acad. Sci. USA*. **109**, 13248-53 (2012).
- [5] Engl C, Schaefer J, Kotta-Loizou I, Buck M. Cellular and molecular phenotypes depending upon the RNA repair system RtcAB of *Escherichia coli*. *Nucleic Acids Res*. 2016 Nov 16;44(20):9933-9941.
- [6] Unciuleac M.C, Goldgur Y, Shuman S. Structure and two-metal mechanism of a eukaryal nick-sealing RNA ligase. *Proc. Natl. Acad. Sci. USA*. **112**, 13868-73 (2015).
- [7] Bechhofer D.H, Deutscher M.P. Bacterial ribonucleases and their roles in RNA metabolism. *Crit. Rev. Biochem. Mol. Biol*. **54**, 242-300 (2019).
- [8] Główny D, Pianka D, Sulej AA, Kozłowski Ł.P, Czarnańska J, Chojnowski G, Skowronek K.J, Bujnicki J.M. Sequence-specific cleavage of dsRNA by Mini-III RNase. *Nucleic Acids Res*. **43**, 2864-73 (2015).

2. Interestingly, the authors did not observe any circular sRNA in *B. subtilis* 168. I propose they express *DucS* from a plasmid in *B. subtilis* and analyze in Northern blots if this sRNA is also present in a circular form or only in a linear form. The latter would indicate that enzymes for circularization are different in both species and, at the same time, corroborate that self-splicing can be excluded. However, if *DucS* is also present in a circular conformation, it would make the search for responsible enzymes much easier, because *liqA* and *liqB* mutants as well as RNase knockout or knockdown strains are available for *B. subtilis*.

Response: Thank you very much for your constructive suggestion. Accordingly, we expressed *DucS* from a plasmid (pBa.*DucS*) in *B. subtilis* (lacking *DucS* homolog) and the result showed that *DucS* could not be circularized in *B. subtilis* (Extended Data Fig. 2a). In addition, since the SRP RNA was highly conserved in *Bacillus* and could be circularized in *B. altitudinis* but not in *B. subtilis*, we also overexpressed SRP RNA of *B. altitudinis* (pBa.SRP) and of *B. subtilis* (pBs.SRP) in both host strains respectively to test the production of circular RNA. Since both hosts harboring chromosome-encoding SRP RNA, we can only infer the expression of plasmid-borne *DucS* from the intensity of hybrid bands. The result showed that SRP RNA from *B. altitudinis* could not be circularized in *B. subtilis*, and conversely, SRP RNA from *B. subtilis* could generate circular RNA in *B. altitudinis*

(Extended Data Fig. 2b). To make the results more convincing, we constructed a recombinant SRP RNA (pBs. rSRP or pBa. rSRP), by inserting an extra sequence (66 bp) into plasmid-borne SRP RNA, to make it be discriminated from chromosomal version. The results further confirmed the above conclusion. Therefore, the existing data suggest that some enzymes for RNA circularization (at least for DucS and SRP RNA) are different in two species, or specific signal exists only in *B. altitudinis*. The comparison of two genomes in-depth will help to reveal crucial differences related to RNA circularization between them.

Back to the first question, for the homologs of Pnkp, it is worth noting that *B. paralicheniformis*, in which SRP RNA can also be circularized, contains two homologs similar with that of *B. altitudinis*, but *B. subtilis* contains only one homolog, PrpE. Thus, difference of the putative RNA repair complex among these species may be the clue for further studies.

Extended Data Fig. 2 Analysis of expression pattern of DucS and SRP RNA in *B. altitudinis* and *B. subtilis*. **a**, DucS from *B. altitudinis* was constructed into a plasmid (pBa.DucS) and transformed into *B. altitudinis* and *B. subtilis*, respectively. Northern blot was used to detect DucS expression pattern in two strains, respectively. 2 μ g total RNA was loaded from indicated strains from cultures at 4, 8 and 24 h. **b**, Plasmid harboring SRP RNA from *B. altitudinis* (pBa.SRP) or from *B. subtilis* (pBs.SRP) was constructed and transformed into *B. altitudinis* and *B. subtilis*, respectively. Empty vector (pEV) as negative control. Northern blot to detect SRP expression pattern in *B. altitudinis* (left) and *B. subtilis* (right), respectively. **c**, Plasmid harboring recombinant SRP RNAs (pBs. rSRP or pBa. rSRP) was constructed by inserting an extra sequence (66 bp) into SRP RNA, and transformed into *B. altitudinis* and *B. subtilis*, respectively. L and C indicate the endogenous linear and circular SRP RNAs. rL and rC indicate the recombinant linear and circular RNAs with an extra sequence. 500 ng total RNA was loaded from indicated strains from cultures at different time points. EB-stained 16S rRNA was used as loading controls for northern blot.

3. The authors observe that the percentage of circular DucS increases over growth. However, what is the signal/trigger for circularization? Most probably, the signal is somehow linked to the expression of an enzyme/protein involved in circularization.

Response: thank you for your very professional suggestion. Since the generation of circular DucS was related to growth phases, appearing at mid-logarithmic phase (8 h) and increasing gradually thereafter, we had thought that nutrition stress might make point. However, we found that the expression pattern of circular DucS from M9 minimal medium, although showing a globally lower abundance, was almost consistent with that cultured in rich LB medium (Extended Data Fig. 3). Actually, we also evaluated the expression pattern of DucS under stresses of NaCl, pH and high temperature (50 °C), and didn't get any significant differences. The result suggests that circularization of DucS may have other triggers which need to be studied in depth in the future. As the reviewer indicated, the signal is somehow linked to the expression of an enzyme/protein involved in circularization. It seems that RNA ligase is a good candidate. The RNA repair operons, *rtcBA* of *E. coli* and *rsr-yrjBA-rtcBA* of *S. Typhimurium*, are both controlled by a σ^{54} -dependent promoter and an activator protein encoded by *rtcR* [1,2]. However, there are significant differences between *E. coli* and *S. Typhimurium* regarding their cellular responses to different types of stress, resulting in differential expression of their respective RNA repair systems. Then, as long as the RNA ligase involved in circularization in *B. altitudinis* has been identified, we will characterize the signal/trigger for activation of this gene/operon.

Extended Data Fig. 3 Northern blots to detect DucS expression in *B. altitudinis* cultured in rich LB medium and M9 minimal medium. Total RNA was extracted from indicated cultures at 4, 8, 12 and 24 h.

Reference:

- [1] Engl C, Schaefer J, Kotta-Loizou I, Buck M. Cellular and molecular phenotypes depending upon the RNA repair system RtcAB of *Escherichia coli*. *Nucleic Acids Res.* **44**, 9933-9941 (2016).
- [2] Kurasz J.E, Hartman C.E, Samuels D.J, Mohanty B.K, Deleveaux A, Mrázek J, Karls A.C. Genotoxic, Metabolic, and Oxidative Stresses Regulate the RNA Repair Operon of *Salmonella enterica* Serovar *Typhimurium*. *J Bacteriol.* **200**, e00476-18 (2018).

4. How can the circular conformation of the same sRNA promote degradation of the linear conformation? Do the authors have any hypothesis or possible explanation?

Response: Thank you for your question. Another reviewer and editor also mentioned that the evidence supporting a biological function of circular form for degradation of the cognate linear form was weak and/or the effect was modest. We agree with this opinion. Even though we observed that the half-lives of linear RNAs decreased significantly in the presence of circular RNAs, we could not come to the conclusion that DucS circular RNAs involved in degradation of linear cognates. Correlation isn't causation. The intrinsic mechanism needs to be revealed. However, we can make a hypothesis: RNA-binding proteins (RBPs) can directly compete with endoribonuclease for the same site to stabilize RNA, or bind to RNA 3' ends to prevent exoribonucleolytic decay [1]. Circular RNA, owing to its different structure, possesses a higher affinity with such RBPs. Then the occurrence of circular RNA can sequester the RBPs, even if not all, and indirectly promote the decay of its linear cognate by ribonuclease. Of course, this is only a hypothesis, and more in-depth researches are needed to reveal the function of circular RNA.

Reference

[1] Holmqvist E, Vogel J. RNA-binding proteins in bacteria. *Nat. Rev. Microbiol.* **16**, 601-615 (2018).

Minor comments:

1. Just a question: The authors used denaturing PAA gels and detected that circular DucS run slower than linear. Did they also take a look in agarose gels? There, circular RNA runs much faster than linear one.

Response: Actually, we initially used agarose gels with denaturant formaldehyde to detect the expression of DucS and observed only one hybridization band at different time points (Extended Data Fig. 4). To figure out the exact size of this band, we switched to denaturing PAA gels and found that there were at least four bands. Therefore, this discovery is a beautiful accident.

Extended Data Fig. 4 Analysis of DucS expression using formaldehyde-denaturing agarose gels. Total RNA was extracted from wildtype *B. altitudinis* SCU11 and 10 μ g total RNA was loaded from cultures at indicated time points.

2. Not only a, b, c, d and e used for labelling the Figures, but also the letter size used for labelling within the figures has to be increased. For example, the subscripted letters in Fig. 3 b, c, d and e are barely decipherable. A possible solution to this problem would be to use no subscript, but simply DucS Δ 16 etc.

Response: Thank you for your kindly advise. We have modified and increased letter size in all figures.

3. Line 77 reference 21 This reference is about Bacillus pumilus, not B. altitudinis

Response: Thank you for your careful work and we are sorry for our carelessness. We should provide clear explanation for the relationship of *Bacillus altitudinis* SCU11, *Bacillus pumilus* SCU11 and BA06. In our previous work, *B. pumilus* SCU11 was obtained by combined mutagenesis from the parent strain BA06, which was classified as *Bacillus pumilus* based mainly on 16S rDNA sequence [1]. But recently the strain has been reclassified as *Bacillus altitudinis* by NCBI database, based on phylogenomics and comparative genomic analyses. Thus, we used *Bacillus altitudinis* instead of *Bacillus pumilus* in this study. To avoid confusing, we provided the explanation for *Bacillus altitudinis* SCU11 in the revised manuscript like this:

“To explore the post-transcriptional regulation in *Bacillus* strains, we previously identified a set of sRNAs in *Bacillus altitudinis* SCU11 (initially classified as *Bacillus pumilus*), a productive strain of extracellular alkaline protease, through an RNA-seq-based approach and subsequent validation by northern blotting”. (Line 70-74)

Reference

[1] Wang, H.Y. et al. Screening and mutagenesis of a novel *Bacillus pumilus* strain producing alkaline protease for dehairing. *Lett. Appl. Microbiol.* **44**, 1-6 (2007).

4. Line 110 reference 22. Is this the correct reference?

Response: Thank you for your careful work. We have changed the reference to the original article that firstly developed this method. (Line 107)

5. Line 211 The references 32,33 are not only for CopraRNA (this is ref. 32), but also for another program, GLASSgo (ref. 33), please correct.

Response: We are sorry for our carelessness. We have rewritten this sentence like this: “The CopraRNA, in combination with GLASSgo, were applied to target prediction, and the top 5 candidates (Supplementary Table 1) were used for electrophoretic mobility shift assays (EMSAs) to test the interactions between DucS and candidate RNAs”. (Line 210-213)

6. The English needs to be corrected by a professional language service. I provide a few examples: Manuscript title: It must read: "involved in" or "is involved in" and not "involves"
line 92 must be corrected into: A DucS knockout strain..was constructed through the CRISPR-Cas9 system and the complementation strain was.....plasmidborne DucS into the Δ ducS strain
line 116 not "slowly shifting transcripts", but "slowly migrating transcripts"
line 116/117 not "made us confusing" but "confused us"
line 131 Correct: " In view of band1 and band2 appeared in 8 h sample" into "Concerning band 1 and band 2 that appeared in the 8 h sample...."

Response: Thank you for your careful work. The language had been revised by a native English-speaking colleague (American Journal Experts) to correct grammatical errors and improve the readability of the manuscript.

Reviewer #3

Comments: This manuscript characterizes a new functional circular RNA in *Bacillus altitudinis* and indicates that a number of other circular RNAs exist in bacteria. The experiments are clear, well controlled, and convincing. The work should spawn others to look for and characterize bacterial circular RNAs and thus the manuscript represents a critical starting point for the field. I have several suggestions for clarifying the results.

1. There are a number of places where important control data are not shown. The data on Line 134 should be shown to confirm the authors' interpretation regarding the 3' end, the data in Line 165 need to be shown to prove that RNase R is behaving as expected, the data in Line 223 should be shown to prove that DucS does not activate all transgenes, and the RT-PCR sequencing data should be shown in Line 304.

Response: Thank you for your careful work and professional comments. These important control data have been added in corresponding Figures and supplementary Figures. Data in Line 133 have been shown in Supplementary Fig. 2c. Data in Line 165 have been shown in Fig. 2c and data in Line 225 have been shown in Supplementary Fig. 6a. RT-qPCR sequence data of Line 312 have been shown in Supplementary Fig. 10a.

2. Fig 4f: RNA half-lives should be more accurately determined rather than simply writing >32 as has been done in many cases.

Response: Thank you for your suggestion. Northern blots data showed that linear RNA levels in strain $\Delta ducS_pDucS$ (at 6 h) were reduced to less than half when rifampicin treatment reached to 32 min, while the linear RNA levels in circular RNA deficiency mutant were almost unchanged. The results clearly display the difference of half-life in two cases. It is inaccurate to calculate the half-life for samples $t_{1/2} > 32$ based on the existing results. The appropriate method is to extend the rifampicin treatment time until the band intensity decreases to lower than half. We will do that when needed.

3. The manuscript should be thoroughly checked for grammatical errors. For example,

“involves” in the title (Line 2) should be “involved”.

Response: We acknowledge your careful work. The language had been revised by a native English-speaking colleague (American Journal Experts) to improve the readability of the manuscript.

4. *A supplemental table with the sequences of all circRNA candidates and the number of detected sequencing reads for each (Fig 5) should be provided.*

Response: Thank you for your careful work and professional comments. Sequences of all circular candidates and the number of detected sequencing reads have been listed in Supplementary Tables 2 and 3.

Minor points:

1. Line 62: Circular RNAs in eukaryotes were discovered more than 30 years ago, e.g. Nigro et al (1991) Cell “Scrambled exons” and Capel et al (1993) Cell “Circular transcripts of the testis-determining gene Sry in adult mouse testis”.

Response: Thank you for your careful work. We are sorry for our carelessness. We have rewritten this sentence like this:

“Circular RNAs, a class of noncoding RNAs first found in eukaryotes more than 40 years ago, [1], have recently been shown to exert biological functions by acting as decoys, transporters or scaffolds [2].” (Line 58-60)

Reference:

[1] Hsu, M.-T. & Coca-Prados, M. Electron microscopic evidence for the circular form of RNA in the cytoplasm of eukaryotic cells. *Nature* **280**, 339-340 (1979).

[2] Kristensen, L.S. et al. The biogenesis, biology and characterization of circular RNAs. *Nat. Rev. Genet.* **20**, 675-691 (2019).

2. Figures 4b, 4c, S7 should be quantified.

Response: Thank you for your careful work and professional comments. Quantitative results have been supplemented in Fig. 4b, 4c and Supplementary Fig. 7.

3. Fig 5d, Supp Fig 10: It would be good to also use a Northern probe that spans the junction as one additional way of proving that at least some of the identified transcripts

are circular RNAs.

Response: Thank you for your professional comments. According to your suggestion, we selected three transcripts with relatively higher abundance of circular forms (SRP RNA, EQK04_RS04100 and Unknown-RNA1) from five identified candidates to carry out the hybridization by junction-spanning probes, and confirmed their circular forms. The results have been added in Supplementary Fig. 10c in the revised version, as follows.

Extended Data Fig. 5 Circular RNAs of SRP RNA, Unknown-RNA1 and EQK04_RS04100 were detected by junction-spanning oligonucleotide probes (C-probe). Probes detecting both linear and circular forms (L/C-probe) were used as control. The junction-spanning probes correspond to 15 nucleotides on each side of junctions of SRP RNA and Unknown-RNA1, and 20 nucleotides on each side of junction of EQK04_RS04100, respectively. RNA (24 h) pre-treated with or without RNase R was used for Northern blots. EB-stained rRNA acted as RNase R digestion control.

4. Line 354: It is known that RNase R does not digest all linear RNAs (PMID: 28444238, 31269210) so this may be the reason why some transcripts appear circular by RNase R Northern blots but not by RT-PCR.

Response: Thank you for your careful work and professional comments. Indeed, RNase R can digest almost all linear RNAs, but it is hard to work on linear RNAs

containing G-quadruplexes or structured 3' ends ^[1]. In addition, bacteria have rho-independent transcriptional terminator which may be one of the reasons why some RNAs are difficult to be digested by RNase R. Therefore, the identification of a circular RNA cannot be only based on digestion of RNase R, but a combination of multiple methods.

Reference

[1] Xiao M.S., Wilusz J.E. An improved method for circular RNA purification using RNase R that efficiently removes linear RNAs containing G-quadruplexes or structured 3' ends. *Nucleic Acids Res.* **47**, 8755–8769 (2019).

Reviewer #1 (Remarks to the Author):

NCOMMS-23-08914

A linear and circular dual-conformation noncoding RNA involved in oxidative stress tolerance of *Bacillus altitudinis*. He et al. and Wang

In their revised manuscript, the authors have acknowledged the importance of including a structural model for DucS RNA termini in understanding potential mechanisms of circularization. However, the model they have included (Supplementary Fig. 11) is not the one that was suggested in my original comments. The secondary structure of the 5' and 3' termini in Supplementary Fig. 11 is almost certain to be incorrect, based on the authors' own data and on the reduced calculated stability of the model. The authors state that RNAfold and Mfold were used to generate the DucS structure model, but neither of these tools is able to predict pseudoknots, and so Supplementary Fig. 11 lacks the highly stable pseudoknotted RNA structure that was outlined in detail in two separate diagrams in my original comments.

Given the authors' stated recognition that the secondary structure adopted by the 5' and 3' termini of DucS informs our understanding of the circularization mechanism, it is puzzling to me that the (currently incorrect) model is not mentioned at all in the Results section. Instead, it appears as the very last Supplementary figure, and is cited for the first time in the Discussion. It would seem more appropriate to refer to the structure model in the Results section titled "Identification of critical sequences for DucS circularization". After all, this is the section in which the authors provide much of the evidence that is consistent with the pseudoknotted structure of the DucS termini. Also, related to this point, it would seem to make sense to incorporate the structure model into main text figures. Currently, the effects of mutations and deletions on circularization efficiency are depicted using graphics devoid of structural information (for example, see Figure 3, in which there is no indication whatsoever of the pseudoknotted base-pairing interactions between the termini). Including a structural model in these main text figures would provide an informative framework that would make it much easier for readers to understand the data in the pertinent context.

The authors have tempered their claims concerning the biological role of circular DucS, as is appropriate. Still, the overarching question that remains is whether the collection of circular RNAs identified in this report have biological importance. The authors seem to argue in the affirmative, in part because levels of circular DucS and SRP RNAs on northern blots increase slightly at later time points (Figures 1b and 5d). But this would also be true if the circularization were catalyzed fortuitously by an undefined ligase. And for SRP RNA, it is not only the circular but also the linear form that accumulates with time (Figure 5d).

As stated in my original comments, the simplest and most likely explanation for the existence of DucS in circular form is the complementarity (19 bp in total) between sequences near the 5' and 3' termini. As demonstrated experimentally by the authors, other such RNAs whose ends are known to be anchored in proximity via base pairing (SRP RNA, tmRNA, RNase P RNA) are also detected in circular form. Crucially, this occurs only in *B. altitudinis*. In their rebuttal letter, the authors show that DucS and SRP RNA (from both *B. altitudinis* and *B. subtilis*) undergo circularization only in *B. altitudinis* and not in *B. subtilis* (Extended Data Figure 2). This reinforces the point that *B. altitudinis* contains a ligase activity that acts generally and nonspecifically upon all RNAs whose ends are brought together by base pairing. There is certainly no known role for the circularization of the well studied RNase P RNA, tmRNA and SRP RNA molecules. These highly structured RNAs are known to be quite stable in their linear forms, and so it is quite unlikely that circularization would provide any benefit in this regard. Of particular note, tmRNA would be rendered non-functional by circularization, as its 3' end must be aminoacylated to interact productively with stalled ribosomes. Thus, one would have to invoke the convoluted hypothesis that in *B. altitudinis* (and only in *B. altitudinis*), circularization imparts added functionality to certain RNAs (DucS, RNaseP, SRP RNA) while abrogating functionality in others (tmRNA). The far more likely explanation, which is suggested by the authors' own data, is that the circularization of the non-conserved DucS RNA results from a cross-reacting ligase activity in *B. altitudinis* that bears no biological relevance.

Reviewer #2 (Remarks to the Author):

In their revised manuscript, the authors performed experiments related to two of my questions .

1) I proposed to look if DucS is also circular in *B. subtilis* and they show now that circularization of DucS is specific for *B. altitudinis* and does not occur in *B. subtilis*.

This is an unexpected finding and suggests that *B. subtilis* and *B. altitudinis* differ in one or several enzymes which I would have thought to be conserved.

2) Since the RNA ligases and the RNases except RNase III are not yet known/investigated in this strain, they could not

further look for enzymes involved in circularization. As one could expect, RNase III has no effect on the linear or circ form of DucS.

3) My last question was what triggers circularization, and the authors tried to find environmental/stress conditions, under which circularization might perhaps be triggered but could not find any.

The only thing that I do not understand is that SRP, the RNA component of the signal recognition particle, is circular in their experiments, I always believed it is a linear RNA.

What the authors should leave out from the paper are the structure predictions of the sRNAs: We have mapped experimentally a lot of sRNA secondary structures, and none of them coincides with one of the 10 predictions you obtain by M-fold. Either the authors map experimentally the secondary structures of the linear and the circular form of DucS (which would show if in the circular form, the target-binding regions are accessible) with single- and double-strand-specific RNases like T1, T2, A and V1 or chemical probes or they omit speculations on hypothetical structures. All prediction programs make researchers believe that an RNA is circular by presenting them with the 5' and the 3' end in close vicinity, but in reality these ends are rarely adjacent.

Although the role and function of this circular RNA and its genesis remain open - due to little information on *B. altitudinis* - the revised version of the manuscript is of interest to the reader of Nature Communications and can be accepted.

Reviewer #3 (Remarks to the Author):

The authors have appropriately addressed my prior comments.

I suggest they add some of the extended data figures that they used in the response to Reviewer 2 to the finalized manuscript. Readers make be interested in those results.

Text clarifications:

Line 58: Some circRNAs in eukaryotes are protein-coding.

Line 162: What linear reference mRNA was examined?

A linear and circular dual-conformation noncoding RNA involved in oxidative stress tolerance
in *Bacillus altitudinis*

(NCOMMS-23-08914A)

Response to the reviewers' comments

Dear editors and reviewers,

Thank you for providing us this opportunity to further revise our manuscript. We also appreciate the thoughtful and constructive comments from the editors and reviewers. We have made corresponding revisions in our manuscript, showing the main changes in the red text. We have copied and pasted all referees' comments below, and addressed each one individually. The original comments are in italic with black color, and the responses are in normal font with blue color. As you will see, we have made every attempt to incorporate these suggestions as thoroughly as possible.

Point-by-point responses to the Reviewers' comments

Reviewer #1:

Comments: *A linear and circular dual-conformation noncoding RNA involved in oxidative stress tolerance of Bacillus altitudinis. He et al. and Wang*

In their revised manuscript, the authors have acknowledged the importance of including a structural model for DucS RNA termini in understanding potential mechanisms of circularization. However, the model they have included (Supplementary Fig. 11) is not the one that was suggested in my original comments. The secondary structure of the 5' and 3' termini in Supplementary Fig. 11 is almost certain to be incorrect, based on the authors' own data and on the reduced calculated stability of the model. The authors state that RNAfold and Mfold were used to generate the DucS structure model, but neither of these tools is able to predict pseudoknots, and so Supplementary Fig. 11 lacks the highly stable pseudoknotted RNA structure that was outlined in detail in two separate diagrams in my original comments.

Given the authors' stated recognition that the secondary structure adopted by the 5' and 3' termini of DucS informs our understanding of the circularization mechanism, it is puzzling to me that the (currently incorrect) model is not mentioned at all in the Results section. Instead, it appears as the very last Supplementary figure, and is cited for the first time in the Discussion. It would seem more appropriate to refer to the structure model in the Results section titled "Identification of critical sequences for DucS circularization". After all, this is the section in which the authors provide much of the evidence that is consistent with the pseudoknotted structure of the DucS termini. Also, related to this point, it would seem to make sense to incorporate the structure model into main text figures. Currently, the effects of mutations and deletions on circularization efficiency are depicted using graphics devoid of structural information (for example, see Figure 3, in which there is no indication whatsoever of the pseudoknotted base-pairing interactions between the termini). Including a structural model in these main text figures would provide an informative framework that would make it much easier for readers to understand the data in the pertinent context.

The authors have tempered their claims concerning the biological role of circular DucS, as is appropriate. Still, the overarching question that remains is whether the collection of circular RNAs identified in this report have biological importance. The authors seem to argue in the affirmative, in part because levels of circular DucS and SRP RNAs on northern blots increase slightly at later time points (Figures 1b and 5d). But this would also be true if the circularization were catalyzed fortuitously by an undefined ligase. And for SRP RNA, it is not only the circular but also the linear form that accumulates with time (Figure 5d).

*As stated in my original comments, the simplest and most likely explanation for the existence of DucS in circular form is the complementarity (19 bp in total) between sequences near the 5' and 3' termini. As demonstrated experimentally by the authors, other such RNAs whose ends are known to be anchored in proximity via base pairing (SRP RNA, tmRNA, RNase P RNA) are also detected in circular form. Crucially, this occurs only in *B. altitudinis*. In their rebuttal letter, the authors show that DucS and SRP RNA (from both *B. altitudinis* and *B. subtilis*) undergo circularization only in *B. altitudinis* and not in *B. subtilis* (Extended Data Figure 2). This reinforces the point that *B. altitudinis* contains a ligase activity that acts generally and nonspecifically upon all RNAs whose ends are brought together by base pairing. There is certainly no known role for the circularization of the well studied RNase P RNA, tmRNA and SRP RNA molecules. These highly structured RNAs are known to be quite stable in their linear forms, and so it is quite unlikely that circularization would provide any benefit in this regard. Of particular note, tmRNA would be rendered non-functional by circularization, as its 3' end must be aminoacylated to interact productively with stalled ribosomes. Thus, one would have to invoke the convoluted hypothesis that in *B. altitudinis* (and only in *B. altitudinis*), circularization imparts added functionality to certain RNAs (DucS, RNaseP, SRP RNA) while abrogating functionality in others (tmRNA). The far more likely explanation, which is suggested by the authors' own data, is that the circularization of the non-conserved DucS RNA results from a cross-reacting ligase activity in *B. altitudinis* that bears no biological*

relevance.

Response: First, we must admit that we do not have a clear understanding of some important characteristics of RNA structure, such as pseudoknots, noncanonical base pairs as well as triplet interactions. We do apologize for failing to understand the meaning of reviewer's prediction; And thus adopted the structure predicted by RNAfold which is unable to predict pseudoknots. Now we realize that pseudoknots are the most prevalent RNA structures with diverse functions, and a number of approaches have been proposed successively to predict secondary structure with pseudoknots, for example, pknotsRG, Probknot, IPknot, SPOTRNA, E2Efold, ATTFold, REDfold, etc. However, base pairs associated with pseudoknots are challenging for both folding-based and machine-learning-based approaches because they are often associated with tertiary interactions that are difficult to predict ^[1]. We are learning how to choose and use these tools, and how to deal with the possible varying results. We hope to obtain helpful information for subsequent experiment. In general, the confidence level in an RNA secondary structure model depends on the number of pieces of computational and experimental information used to predict that model, and a true secondary structure can be verified when the 3D structure is determined ^[2]. Based on this consideration, as well as the suggestions from another Reviewer and the Editor, we have deleted the content about secondary structure prediction.

So far as the potential function of circular RNA, we do not have directed evidence at present to support the biological role of circular DucS or SRP RNA. We also agree with your comment that there is no known role for the circularization of the well-studied RNase P RNA, tmRNA and SRP RNA molecules, and we are unable here to provide a reasonable hypothesis about the potential function of their circular forms. We have to admit that it is a limitation of this study. Of course, we will focus on functional characterization subsequently. It seems that the CRISPR/Cas13d-mediated circular RNA knockdown is a feasible approach for this study ^[3], ^[4]. We hope the function of bacterial circular RNAs, perhaps DucS or SRP RNA, perhaps others, will be revealed in the future.

Reference:

[1] Singh J, Hanson J, Paliwal K, Zhou Y. RNA secondary structure prediction using an ensemble of two-dimensional deep neural networks and transfer learning. *Nat Commun.* 2019 Nov 27;10(1):5407.

[2] Vicens Q, Kieft JS. Thoughts on how to think (and talk) about RNA structure. *Proc Natl Acad Sci U S A.* 2022 Apr 26; 119(17):e2112677119.

[3] Li S, Wu H, Chen LL. Screening circular RNAs with functional potential using the RfxCas13d/BSJ-gRNA system. *Nat Protoc.* 2022 Sep;17(9):2085-2107.

[4] Zhang K, Zhang Z, Kang J, Chen J, Liu J, Gao N, Fan L, Zheng P, Wang Y, Sun J. CRISPR/Cas13d-Mediated Microbial RNA Knockdown. *Front Bioeng Biotechnol.* 2020 Jul 30;8:856.

Reviewer #2:

Comments: In their revised manuscript, the authors performed experiments related to two of my questions.

1. I proposed to look if DucS is also circular in *B. subtilis* and they show now that circularization of DucS is specific for *B. altitudinis* and does not occur in *B. subtilis*.

This is an unexpected finding and suggests that *B. subtilis* and *B. altitudinis* differ in one or several enzymes which I would have thought to be conserved.

Response: Thank you for your prior suggestion. Based on the experiments, we agree with your opinion that *B. subtilis* and *B. altitudinis* differ in one or several enzymes related to RNA circularization.

2. Since the RNA ligases and the RNases except RNase III are not yet known/investigated in this strain, they could not further look for enzymes involved in circularization. As one could expect, RNase III has no effect on the linear or circ form of DucS.

Response: Although the genome annotation of *B. altitudinis* is incomplete, we have found a few candidates for the RNA ligases and the RNases. The knockout strains are under construction.

3. My last question was what triggers circularization, and the authors tried to find environmental/stress conditions, under which circularization might perhaps be triggered but could not find any.

Response: Although the triggers for circularization failed to be revealed in the previous experiment, we will characterize the signal/trigger when the critical enzyme/protein

involved in circularization has been identified, by focusing on the activator of this gene/operon.

4. The only thing that I do not understand is that SRP, the RNA component of the signal recognition particle, is circular in their experiments, I always believed it is a linear RNA.

Response: We want to emphasize that, even though circular SRP RNA exists in *B. altitudinis* and *B. paralicheniformis*, the linear form is absolutely prevalent in both strains (Fig. 5d).

5. What the authors should leave out from the paper are the structure predictions of the sRNAs: We have mapped experimentally a lot of sRNA secondary structures, and none of them coincides with one of the 10 predictions you obtain by M-fold. Either the authors map experimentally the secondary structures of the linear and the circular form of DucS (which would show if in the circular form, the target-binding regions are accessible) with single- and double-strand-specific RNases like T1, T2, A and V1 or chemical probes or they omit speculations on hypothetical structures. All prediction programs make researchers believe that an RNA is circular by presenting them with the 5' and the 3' end in close vicinity, but in reality these ends are rarely adjacent.

Response: Following your suggestion, as well as the Editor's, we have deleted the content about secondary structure prediction.

Reviewer #3:

Comments: The authors have appropriately addressed my prior comments.

1. I suggest they add some of the extended data figures that they used in the response to Reviewer 2 to the finalized manuscript. Readers make be interested in those results.

Response: The data about the expression of DucS and SRP RNA in heterologous hosts is an interesting but preliminary finding. According to the Editor's suggestion, it is currently unsuitable to be added.

2. Line 58: Some circRNAs in eukaryotes are protein-coding.

Response: We have added the function of protein-coding in the sentence like this:

“Circular RNAs, a class of noncoding RNAs first found in eukaryotes more than 40 years ago, have recently been shown to exert biological functions by acting as decoys, transporters, scaffolds, templates for translation, etc.” (Line 62-64)

3. Line 162: What linear reference mRNA was examined?

Response: Thank you for your careful work. The linear reference mRNA comes from *yaaA* (WP_008347598.1).